# Universal Few-shot Learning of Dense Prediction Tasks with Visual Token Matching

**Donggyun Kim**[1]**, Jinwoo Kim**[1]**, Seongwoong Cho**[1]**, Chong Luo**[2]**, Seunghoon Hong**[1]

[1] School of Computing, KAIST
[2] Microsoft Research Asia

`{kdgyun425, jinwoo-kim, seongwoongjo, seunghoon.hong}@kaist.ac.kr,`
`chong.luo@microsoft.com`

## Abstract

Dense prediction tasks are a fundamental class of problems in computer vision. As supervised methods suffer from high pixel-wise labeling cost, a few-shot learning solution that can learn any dense task from a few labeled images is desired. Yet, current few-shot learning methods target a restricted set of tasks such as semantic segmentation, presumably due to challenges in designing a general and unified model that is able to flexibly and efficiently adapt to arbitrary tasks of unseen semantics. We propose Visual Token Matching (VTM), a universal few-shot learner for arbitrary dense prediction tasks. It employs non-parametric matching on patch-level embedded tokens of images and labels that encapsulates all tasks. Also, VTM flexibly adapts to any task with a tiny amount of task-specific parameters that modulate the matching algorithm. We implement VTM as a powerful hierarchical encoder-decoder architecture involving ViT backbones where token matching is performed at multiple feature hierarchies. We experiment VTM on a challenging variant of Taskonomy dataset and observe that it robustly few-shot learns various unseen dense prediction tasks. Surprisingly, it is competitive with fully supervised baselines using only 10 labeled examples of novel tasks (0.004% of full supervision) and sometimes outperforms using 0.1% of full supervision. Codes are available at `https://github.com/GitGyun/visual_token_matching`.

## 1 Introduction

Dense prediction tasks constitute a fundamental class of computer vision problems, where the goal is to learn a mapping from an input image to a pixel-wise annotated label. Examples include semantic segmentation, depth estimation, edge detection, and keypoint detection, to name a few (Zamir et al., 2018; Cai & Pu, 2019). While supervised methods achieved remarkable progress, they require a substantial amount of manually annotated pixel-wise labels, leading to a massive and often prohibitive per-task labeling cost (Kang et al., 2019; Liu et al., 2020; Ouali et al., 2020). Prior work involving transfer and multi-task learning have made efforts to generally relieve the burden, but they often assume that relations between tasks are known in advance, and still require a fairly large amount of labeled images of the task of interest (*e.g.*, thousands) (Zamir et al., 2018; Standley et al., 2020; O Pinheiro et al., 2020; Wang et al., 2021). This motivates us to seek a *few-shot learning* solution that can *universally* **learn arbitrary dense prediction tasks from** *a few* (*e.g.*, ten) **labeled images**.

However, existing few-shot learning methods for computer vision are specifically targeted to solve a restricted set of tasks, such as classification, object detection, and semantic segmentation (Vinyals et al., 2016; Kang et al., 2019; Min et al., 2021). As a result, they often exploit prior knowledge and assumptions specific to these tasks in designing model architecture and training procedure, therefore not suited for generalizing to arbitrary dense prediction tasks (Snell et al., 2017; Fan et al., 2022; Iqbal et al., 2022; Hong et al., 2022). To our knowledge, no prior work in few-shot learning provided approaches to solve arbitrary dense prediction tasks in a universal manner.

We argue that designing a universal few-shot learner for arbitrary dense prediction tasks must meet the following desiderata. First, the learner must have a unified architecture that can handle arbitrary tasks by design, and share most of the parameters across tasks so that it can acquire generalizable knowledge for few-shot learning of arbitrary unseen tasks. Second, the learner should flexibly adapt its prediction mechanism to solve diverse tasks of unseen semantics, while being efficient enough to prevent over-fitting. Designing such a learner is highly challenging, as it should be general and unified while being able to flexibly adapt to any unseen task without over-fitting few examples.

In this work, we propose **Visual Token Matching (VTM)**, a universal few-shot learner for arbitrary dense prediction tasks. We draw inspiration from the cognitive process of analogy making (Mitchell, 2021); given a few examples of a new task, humans can quickly understand how to relate input and output based on a *similarity* between examples (*i.e.*, assign similar outputs to similar inputs), while flexibly changing the notion of similarity to the given context. In VTM, we implement analogy-making for dense prediction as patch-level non-parametric matching, where the model learns the similarity in image patches that captures the similarity in label patches. Given a few labeled examples of a novel task, it first adapts its similarity that describes the given examples well, then predicts the labels of an unseen image by combining the label patches of the examples based on image patch similarity. Despite the simplicity, the model has a unified architecture for arbitrary dense prediction tasks since the matching algorithm encapsulates all tasks and label structures (*e.g.*, continuous or discrete) by nature. Also, we introduce only a small amount of task-specific parameters, which makes our model robust to over-fitting as well as flexible.

Our contributions are as follows. **(1)** For the first time to our knowledge, we propose and tackle the problem of universal few-shot learning of arbitrary dense prediction tasks. We formulate the problem as episodic meta-learning and identify two key desiderata of the learner – unified architecture and adaptation mechanism. **(2)** We propose Visual Token Matching (VTM), a novel universal few-shot learner for dense prediction tasks. It employs non-parametric matching on tokenized image and label embeddings, which flexibly adapts to unseen tasks using a tiny amount of task-specific parameters. **(3)** We implement VTM as a powerful hierarchical encoder-decoder architecture, where token matching is performed at multiple feature hierarchies using attention mechanism. We employ ViT image and label encoders (Dosovitskiy et al., 2020) and a convolutional decoder (Ranftl et al., 2021), which seamlessly works with our algorithm. **(4)** We demonstrate VTM on a challenging variant of Taskonomy dataset (Zamir et al., 2018) and observe that it robustly few-shot learns various unseen dense prediction tasks. Surprisingly, it is competitive to or outperforms fully supervised baselines given extremely few examples ($0.1\%$), **sometimes using only 10 labeled images (**$< 0.004\%$**)**.

## 2 PROBLEM SETUP

We propose and tackle the problem of universal few-shot learning of arbitrary dense prediction tasks. In our setup, we consider any arbitrary task $\mathcal{T}$ that can be expressed as follows:

$$\mathcal{T} : \mathbb{R}^{H \times W \times 3} \to \mathbb{R}^{H \times W \times C_{\mathcal{T}}}, \quad C_{\mathcal{T}} \in \mathbb{N}. \tag{1}$$

This subsumes a wide range of vision tasks including semantic segmentation, depth estimation, surface normal prediction, edge prediction, to name a few, varying in structure of output space, *e.g.*, dimensionality ($C_{\mathcal{T}}$) and topology (discrete or continuous), as well as the required knowledge.

Our goal is to build a universal few-shot learner $\mathcal{F}$ that, for *any* such task $\mathcal{T}$, can produce predictions $\hat{Y}^q$ for an unseen image (*query*) $X^q$ given a few labeled examples (*support set*) $\mathcal{S}_{\mathcal{T}}$:

$$\hat{Y}^q = \mathcal{F}(X^q; \mathcal{S}_{\mathcal{T}}), \quad \mathcal{S}_{\mathcal{T}} = \{(X^i, Y^i)\}_{i \le N}. \tag{2}$$

To build such a universal few-shot learner $\mathcal{F}$, we adopt the conventional episodic training protocol where the training is composed of multiple *episodes*, each simulating a few-shot learning problem. To this end, we utilize a meta-training dataset $\mathcal{D}_{\text{train}}$ that contains labeled examples of diverse dense prediction tasks. Each training episode simulates a few-shot learning scenario of a specific task $\mathcal{T}_{\text{train}}$ in the dataset – the objective is to produce correct labels for query images given a support set. By experiencing multiple episodes of few-shot learning, the model is expected to learn general knowledge for fast and flexible adaptation to novel tasks. At test time, the model is asked to perform few-shot learning on arbitrary *unseen* tasks $\mathcal{T}_{\text{test}}$ not included in the training dataset ($\mathcal{D}_{\text{train}}$).

An immediate challenge in handling arbitrary tasks in Eq. 1 is that each task in both meta-training and testing has different output structures (*i.e.*, output dimension $C_{\mathcal{T}}$ varies per task), making it difficult to design a single, unified parameterization of a model for all tasks. As a simple yet general solution, we cast a task $\mathcal{T} : \mathbb{R}^{H \times W \times 3} \to \mathbb{R}^{H \times W \times C_{\mathcal{T}}}$ into $C_{\mathcal{T}}$ *single-channel* sub-tasks $\mathcal{T}_1, \cdots, \mathcal{T}_{C_{\mathcal{T}}}$ of learning each channel, and model each sub-task $\mathcal{T}_c : \mathbb{R}^{H \times W \times 3} \to \mathbb{R}^{H \times W \times 1}$ independently using the shared model $\mathcal{F}$ in Eq. 2. Although multi-channel information is beneficial in general, we observe that its impact is negligible in practice, while the channel-wise decomposition introduces other useful benefits such as augmenting the number of tasks in meta-training, flexibility to generalize to arbitrary dimension of unseen tasks, and more efficient parameter-sharing within and across tasks. Without loss of generality, the rest of the paper considers that every task is of single-channel label.

## 2.1 CHALLENGES AND DESIDERATA

The above problem setup is universal, potentially benefiting various downstream vision applications. Yet, to our knowledge, no prior work in few-shot learning attempted to solve it. We attribute this to two unique desiderata for a universal few-shot learner that pose a challenge to current methods.

**Task-Agnostic Architecture** As any arbitrary unseen task $\mathcal{T}_{\text{test}}$ can be encountered in test-time, the few-shot learner must have a unified architecture that can handle all dense prediction tasks by design. This means we cannot exploit any kind of prior knowledge or inductive bias specific to certain tasks. For example, we cannot adopt common strategies in few-shot segmentation such as class prototype or binary masking, as they rely on the discrete nature of categorical labels while the label space of $\mathcal{T}_{\text{test}}$ can be arbitrary (continuous or discrete). Ideally, the unified architecture would allow the learner to acquire generalizable knowledge for few-shot learning any unseen tasks, as it enables sharing most of the model parameters across all tasks in meta-training and testing.

**Adaptation Mechanism** On top of the unified architecture, the learner should have a flexible and efficient adaptation mechanism to address highly diverse semantics of the unseen tasks $\mathcal{T}_{\text{test}}$. This is because tasks of different semantics can require distinct sets of *features* – depth estimation requires 3D scene understanding, while edge estimation prefers low-level image gradient. As a result, even with a unified architecture, a learner that depends on a fixed algorithm and features would either underfit to various training tasks in $\mathcal{D}_{\text{train}}$, or fail for an unseen task $\mathcal{T}_{\text{test}}$ that has a completely novel semantics. Thus, our few-shot learner should be flexibly adapt its features to a given task $\mathcal{T}$ based on the support set $\mathcal{S}_{\mathcal{T}}$, *e.g.*, through task-specific parameters. At the same time, the adaptation mechanism should be parameter-efficient (*e.g.*, using a tiny amount of task-specific parameters) to prevent over-fitting to training tasks $\mathcal{T}_{\text{train}}$ or test-time support set $\mathcal{S}_{\mathcal{T}_{\text{test}}}$.

## 3 VISUAL TOKEN MATCHING

We introduce **Visual Token Matching (VTM)**, a universal few-shot learner for arbitrary dense prediction tasks that is designed to flexibly adapt to novel tasks of diverse semantics. We first discuss the underlying motivation of VTM in Section 3.1 and discuss its architecture in Section 3.2.

### 3.1 MOTIVATION

We seek for a general function form of Eq. 2 that produces structured labels of arbitrary tasks with a unified framework. To this end, we opt into a non-parametric approach that operates on patches, where the query label is obtained by weighted combination of support labels. Let $X = \{\mathbf{x}_j\}_{j \leq M}$ denote an image (or label) on patch grid of size $M = h \times w$, where $\mathbf{x}_j$ is $j$-th patch. Given a query image $X^q = \{\mathbf{x}_j^q\}_{j \leq M}$ and a support set $\{(X^i, Y^i)\}_{i \leq N} = \{(\mathbf{x}_k^i, \mathbf{y}_k^i)\}_{k \leq M, i \leq N}$ for a task $\mathcal{T}$, we project all patches to embedding spaces and predict the query label $Y^q = \{\mathbf{y}_j^q\}_{j \leq M}$ patch-wise by,

$$g(\mathbf{y}_j^q) = \sum_{i \leq N} \sum_{k \leq M} \sigma\left(f_{\mathcal{T}}(\mathbf{x}_j^q), f_{\mathcal{T}}(\mathbf{x}_k^i)\right) g(\mathbf{y}_k^i), \tag{3}$$

where $f_{\mathcal{T}}(\mathbf{x}) = f(\mathbf{x}; \theta, \theta_{\mathcal{T}}) \in \mathbb{R}^d$ and $g(\mathbf{y}) = g(\mathbf{y}; \phi) \in \mathbb{R}^d$ correspond to the image and label encoder, respectively, and $\sigma : \mathbb{R}^d \times \mathbb{R}^d \to [0, 1]$ denotes a similarity function defined on the image patch embeddings. Then, each predicted label embedding can be mapped to a label patch $\hat{\mathbf{y}}_j^q = h(g(\mathbf{y}_j^q))$ by introducing a label decoder $h \approx g^{-1}$.

Note that Eq. 3 generalizes the Matching Network (MN) (Vinyals et al., 2016), and serves as a general framework for arbitrary dense prediction tasks[1]. First, while MN interpolates raw categorical labels $\mathbf{y}$ for classification (*i.e.*, $g$ is an identity function), we perform matching on the general embedding space of the label encoder $g(\mathbf{y})$; it encapsulates arbitrary tasks (*e.g.*, discrete or continuous) into the common embedding space, thus enabling the matching to work in a consistent way agnostic to tasks. Second, while MN exploits a fixed similarity of images $\sigma(f(\mathbf{x}^q), f(\mathbf{x}^i))$, we modulate the similarity $\sigma(f_{\mathcal{T}}(\mathbf{x}^q), f_{\mathcal{T}}(\mathbf{x}^i))$ adaptively to any given task by switching the task-specific parameters $\theta_{\mathcal{T}}$. Having adaptable components in the image encoder $f_{\mathcal{T}}$ is essential in our problem since it can adapt the representations to reflect features unique in each task. Finally, our method employs an additional decoder $h$ to project the predicted label embeddings to the output space.

Once trained, the prediction mechanism in Eq. 3 can easily adapt to *unseen* tasks at test-time. Since the label encoder $g$ is shared across tasks, we can use it to embed the label patches of unseen

---

[1]While MN performs image-level classification, we consider its extension to patch-level embeddings.

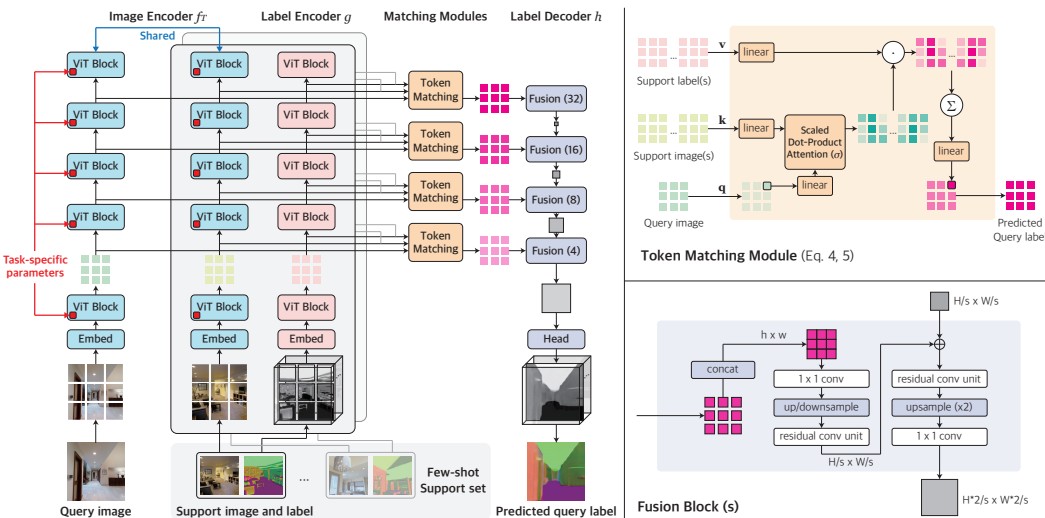

Figure 1: Overall architecture of VTM. Our model is a hierarchical encoder-decoder with four main components: the image encoder $f_\mathcal{T}$, label encoder $g$, label decoder $h$, and the matching module. See the text for more detailed descriptions.

tasks with frozen parameters $\phi$. Adaptation to a novel task is performed by the image encoder $f_\mathcal{T}$, by optimizing the task-specific parameters $\theta_\mathcal{T}$ which take a small portion of the model. This allows our model to robustly adapt to unseen tasks of various semantics with a small support set. In experiments, our model becomes competitive to supervised method using less than 0.1% of labels.

### 3.2 ARCHITECTURE

Figure 1 illustrates our model. Our model has a hierarchical encoder-decoder architecture that implements patch-level non-parametric matching of Eq. 3 in multiple hierarchies with four components: image encoder $f_\mathcal{T}$, label encoder $g$, label decoder $h$, and the matching module. Given the query image and the support set, the image encoder first extracts patch-level embeddings (*tokens*) of each query and support image independently. The label encoder similarly extracts tokens of each support label. Given the tokens at each hierarchy, the matching module performs non-parametric matching of Eq. 3 to infer the tokens of query label, from which the label decoder forms the raw query label.

**Image Encoder**   We employ a Vision Transformer (ViT) (Dosovitskiy et al., 2020) for our image encoder. The ViT is applied to query and each support image independently while sharing weights, which produces tokenized representation of image patches at multiple hierarchies. Similar to Ranftl et al. (2021), we extract tokens at four intermediate ViT blocks to form hierarchical features. To aid learning general representation for a wide range of tasks, we initialize the parameters from pretrained BEiT (Bao et al., 2021), which is self-supervised thus less biased towards specific tasks.

As discussed in Section 3.1, we design the image encoder to have two sets of parameters $\theta$ and $\theta_\mathcal{T}$, where $\theta$ is shared across all tasks and $\theta_\mathcal{T}$ is specific to each task $\mathcal{T}$. Among many candidates to design an adaptation mechanism through $\theta_\mathcal{T}$, we find that *bias tuning* (Cai et al., 2020; Zaken et al., 2022) provides the best efficiency and performance empirically. To this end, we employ separate sets of biases for each task in both meta-training and meta-testing, while sharing all the other parameters.

**Label Encoder**   The label encoder employs the same ViT architecture as the image encoder and extracts token embeddings of the support labels. Similar to the image encoder, the label encoder is applied to each support label independently, with a major difference that it sees one channel at a time since we treat each channel as an independent task as discussed in Section 2. Then the label tokens are extracted from multiple hierarchies that matches the image encoder. Contrary to the image encoder, all parameters of the label encoder are trained from scratch and shared across tasks.

**Matching Module**   We implement the matching module at each hierarchy as a multihead attention layer (Vaswani et al., 2017). At each hierarchy of the image and label encoder, we first obtain the tokens of the query image $X^q$ as $\{\mathbf{q}_j\}_{j \leq M}$ and support set $\{(X^i, Y^i)\}_{i \leq N}$ as $\{(\mathbf{k}_k^i, \mathbf{v}_k^i)\}_{k \leq M, i \leq N}$ from the intermediate layers of image and label encoders, respectively. We then stack the tokens to

row-matrices, $\mathbf{q} \in \mathbb{R}^{M \times d}$ and $\mathbf{k}, \mathbf{v} \in \mathbb{R}^{NM \times d}$. Then, the query label tokens at the hierarchy are inferred as output of a multihead attention layer, as follows:

$$\text{MHA}(\mathbf{q}, \mathbf{k}, \mathbf{v}) = \text{Concat}(\mathbf{o}_1, ..., \mathbf{o}_H) w^O, \tag{4}$$

$$\text{where } \mathbf{o}_h = \text{Softmax}\left(\frac{\mathbf{q}w_h^Q(\mathbf{k}w_h^K)^\top}{\sqrt{d_H}}\right) \mathbf{v}w_h^V, \tag{5}$$

where $H$ is number of heads, $d_H$ is head size, and $w_h^Q, w_h^K, w_h^V \in \mathbb{R}^{d \times d_H}$, $w^O \in \mathbb{R}^{Hd_H \times d}$.

Remarkably, each attention head in Eq. 5 implements the intuition of the non-parametric matching in Eq. 3. This is because each query label token is inferred as a weighted combination of support label tokens $\mathbf{v}$, based on the similarity between the query and support image tokens $\mathbf{q}$, $\mathbf{k}$. Here, the similarity function $\sigma$ of Eq. 3 is implemented as scaled dot-product attention. Since each head involves different trainable projection matrices $w_h^Q, w_h^K, w_h^V$, the multihead attention layer in Eq. 4 is able to learn multiple branches (heads) of matching algorithm with distinct similarity functions.

**Label Decoder** The label decoder $h$ receives the query label tokens inferred at multiple hierarchies, and combines them to predict the query label of original resolution. We adopt the multi-scale decoder architecture of Dense Prediction Transformer (Ranftl et al., 2021) as it seamlessly works with ViT encoders and multi-level tokens. At each hierarchy of the decoder, the inferred query label tokens are first spatially concatenated to a feature map of constant size ($M \to h \times w$). Then, (transposed) convolution layers of different strides are applied to each feature map, producing a feature pyramid of increasing resolution. The multi-scale features are progressively upsampled and fused by convolutional blocks, followed by a convolutional head for final prediction.

Similar to the label encoder, all parameters of the label decoder are trained from scratch and shared across tasks. This lets the decoder to meta-learn a generalizable strategy of decoding a structured label from the predicted query label tokens. Following the channel split in Section 2, the output of the decoder is single-channel, which allows it to be applied to tasks of arbitrary number of channels.

### 3.3 TRAINING AND INFERENCE

We train our model on a labeled dataset $\mathcal{D}_{\text{train}}$ of training tasks $\mathcal{T}_{\text{train}}$ following the standard episodic meta-learning protocol. At each episode of task $\mathcal{T}$, we sample two labeled sets $\mathcal{S}_\mathcal{T}, \mathcal{Q}_\mathcal{T}$ from $\mathcal{D}_{\text{train}}$. Then we train the model to predict labels in $\mathcal{Q}_\mathcal{T}$ using $\mathcal{S}_\mathcal{T}$ as support set. We repeat the episodes with various dense prediction tasks in $\mathcal{D}_{\text{train}}$ so that the model can learn a general knowledge about few-shot learning. Let $\mathcal{F}(X^q; \mathcal{S}_\mathcal{T})$ denote the prediction of the model on $X^q$ using the support set $\mathcal{S}_\mathcal{T}$. Then the model $(f_\mathcal{T}, g, h, \sigma)$ is trained by the following learning objective in end-to-end:

$$\min_{f_\mathcal{T}, g, h, \sigma} \mathbb{E}_{\mathcal{S}_\mathcal{T}, \mathcal{Q}_\mathcal{T} \sim \mathcal{D}_{\text{train}}} \left[ \frac{1}{|\mathcal{Q}_\mathcal{T}|} \sum_{(X^q, Y^q) \in \mathcal{Q}_\mathcal{T}} \mathcal{L}\left(Y^q, \mathcal{F}(X^q; \mathcal{S}_\mathcal{T}))\right) \right], \tag{6}$$

where $\mathcal{L}$ is the loss function. We use cross-entropy loss for semantic segmentation task and $L1$ loss for the others in our experiments. Note that the objective (Eq. 6) does not explicitly enforce the matching equation (Eq. 3) in token space, allowing some knowledge for prediction to be handled by the label decoder $h$, since we found that introducing explicit reconstruction loss on tokens deteriorates the performance in our initial experiments. During training, as we have a fixed number of training tasks $\mathcal{T}_{\text{train}}$, we keep and train separate sets of bias parameters of the image encoder $f_\mathcal{T}$ for each training task (which are assumed to be channel-splitted).

After training on $\mathcal{D}_{\text{train}}$, the model is few-shot evaluated on novel tasks $\mathcal{T}_{\text{test}}$ given a support set $\mathcal{S}_{\mathcal{T}_{\text{test}}}$. We first perform adaptation of the model by fine-tuning bias parameters of the image encoder $f_\mathcal{T}$ using the support set $\mathcal{S}_{\mathcal{T}_{\text{test}}}$. For this, we simulate episodic meta-learning by randomly partitioning the support set into a sub-support set $\tilde{\mathcal{S}}$ and a sub-query set $\tilde{\mathcal{Q}}$, such that $\mathcal{S}_{\mathcal{T}_{\text{test}}} = \tilde{\mathcal{S}} \,\dot\cup\, \tilde{\mathcal{Q}}$.

$$\min_{\theta_\mathcal{T}} \mathbb{E}_{\tilde{\mathcal{S}}, \tilde{\mathcal{Q}} \sim \mathcal{S}_{\mathcal{T}_{\text{test}}}} \left[ \frac{1}{|\tilde{\mathcal{Q}}|} \sum_{(X^q, Y^q) \in \tilde{\mathcal{Q}}} \mathcal{L}\left(Y^q, \mathcal{F}(X^q; \tilde{\mathcal{S}})\right) \right], \tag{7}$$

where $\theta_\mathcal{T}$ denotes bias parameters of the image encoder $f_\mathcal{T}$. The portion of parameters to be fine-tuned is negligible so the model can avoid over-fitting on the small support set $\mathcal{S}_{\mathcal{T}_{\text{test}}}$. After fine-tuned, the model is evaluated by predicting the label of unseen query image using the support set $\mathcal{S}_{\mathcal{T}_{\text{test}}}$.

## 4 RELATED WORK

To the best of our knowledge, the problem of universal few-shot learning of dense prediction tasks remains unexplored. Existing few-shot learning approaches for dense prediction are targeted to specific tasks that require learning unseen *classes* of objects, such as semantic segmentation (Shaban et al., 2017; Wang et al., 2019; Iqbal et al., 2022), instance segmentation (Michaelis et al., 2018; Fan et al., 2020b), and object detection (Fan et al., 2020a; Wang et al., 2020), rather than general tasks. As categorical labels are discrete in nature, most of the methods involve per-class average pooling of support image features, which cannot be generalized to regression tasks as there would be infinitely many "classes" of continuous labels. Others utilize masked correlation between support and query features (Min et al., 2021; Hong et al., 2022), learn a Gaussian Process on features (Johnander et al., 2021), or train a classifier weight prediction model (Kang et al., 2019). In principle, these architectures can be extended to more general dense prediction tasks with slight modification (Section 5), yet their generalizability to unseen dense prediction tasks, rather than classes, has not been explored.

As our method involves task-specific tuning of a small portion of parameters, it is related to transfer learning that aims to efficiently fine-tune a pre-trained model to downstream tasks. In natural language processing (NLP), language models pre-trained on large-scale corpus (Kenton & Toutanova, 2019; Brown et al., 2020) show outstanding performance on downstream tasks with fine-tuning a minimal amount of parameters (Houlsby et al., 2019; Zaken et al., 2022; Lester et al., 2021). Following the emergence of pre-trained Vision Transformers (Dosovitskiy et al., 2020), similar adaptation approaches have been proven successful in the vision domain (Li et al., 2021; Jia et al., 2022; Chen et al., 2022). While these approaches reduce the amount of *parameters* required for state-of-the-art performance on downstream tasks, they still require a large amount of *labeled images* for fine-tuning (*e.g.*, thousands). In this context, our method can be seen as a few-shot extension of the adaptation methods, by incorporating a general few-shot learning framework and a powerful architecture.

## 5 EXPERIMENTS

### 5.1 EXPERIMENTAL SETUP

**Dataset**   We construct a variant of the Taskonomy dataset (Zamir et al., 2018) to simulate few-shot learning of unseen dense prediction tasks. Taskonomy contains indoor images with various annotations, where we choose ten dense prediction tasks of diverse semantics and output dimensions: semantic segmentation (SS), surface normal (SN), Euclidean distance (ED), Z-buffer depth (ZD), texture edge (TE), occlusion edge (OE), 2D keypoints (K2), 3D keypoints (K3), reshading (RS), and principal curvature (PC),[2]. We partition the ten tasks to construct a 5-fold split, in each of which two tasks are used for few-shot evaluation ($\mathcal{T}_{test}$) and the remaining eight are used for training ($\mathcal{T}_{train}$). To perform evaluation on tasks of novel semantics, we carefully construct the partition such that tasks for training and test are sufficiently different from each other *e.g.*, by grouping edge tasks (TE, OE) together as test tasks. The split is shown in Table 1. We process some single-channel tasks (ED, TE, OE) to multiple channels to increase task diversity, and standardize all labels to $[0, 1]$. Additional details are in Appendix A.

**Baselines**   We compare our method (VTM) with two classes of learning approaches.

- **Fully supervised baselines** have an access to the full supervision of test tasks $\mathcal{T}_{test}$ during training, and thus serve as upper bounds of few-shot performance. We consider two state-of-the-art baselines in supervised learning and multi-task learning of general dense prediction tasks – DPT (Ranftl et al., 2021) and InvPT (Ye & Xu, 2022), respectively, where DPT is trained on each single task independently and InvPT is trained jointly on all tasks.

- **Few-shot learning baselines** do not have an access to the test tasks $\mathcal{T}_{test}$ during training, and are given only a few labeled images at the test-time. As there are no prior few-shot method developed for universal dense prediction tasks, we adapt state-of-the-art few-shot segmentation methods to our setup. We choose three methods, DGPNet (Johnander et al., 2021), HSNet (Min et al., 2021), and VAT (Hong et al., 2022), whose architectures are either inherently task-agnostic (DGPNet) or can be simply extended (HSNet, VAT) to handle general label spaces for dense prediction tasks. We describe the modification on HSNet and VAT in Appendix B.

---

[2]We choose all dense prediction tasks defined on RGB images with pixel-wise loss functions.

Table 1: Quantitative comparison on Taskonomy dataset. Few-shot baselines are 10-shot evaluated on each fold after being trained on the tasks from the other folds, where fully-supervised baselines are trained and evaluated on tasks from each fold (DPT) or all folds (InvPT).

| Supervision | Model | Tasks | | | | | | | | | |
| --- | --- | --- | --- | --- | --- | --- | --- | --- | --- | --- | --- |
| | | Fold 1 | | Fold 2 | | Fold 3 | | Fold 4 | | Fold 5 | |
| | | SS | SN | ED | ZD | TE | OE | K2 | K3 | RS | PC |
| | | mIoU ↑ | mErr ↓ | RMSE ↓ | RMSE ↓ | RMSE ↓ | RMSE ↓ | RMSE ↓ | RMSE ↓ | RMSE ↓ | RMSE ↓ |
| Full | DPT | **0.4449** | **6.4414** | **0.0534** | **0.0268** | **0.0188** | **0.0689** | **0.0358** | **0.0357** | **0.0860** | **0.0347** |
| | InvPT | 0.3900 | 12.9249 | 0.0589 | 0.0298 | 0.0517 | 0.0788 | 0.0456 | 0.0384 | 0.0949 | 0.0370 |
| 10-Shot ($< 0.004\%$) | HSNet | 0.1069 | 24.9120 | 0.2375 | 0.0748 | 0.1746 | 0.1643 | 0.1056 | 0.0651 | 0.2627 | 0.0610 |
| | VAT | 0.0353 | 25.8134 | 0.2718 | 0.0779 | 0.1719 | 0.1655 | 0.1450 | 0.0678 | 0.2709 | 0.0796 |
| | DGPNet | 0.0261 | 29.1668 | 0.4579 | 0.2846 | 0.1881 | 0.2130 | 0.1104 | 0.1308 | 0.3680 | 0.3574 |
| | **Ours** | **0.4097** | **11.4391** | **0.0741** | **0.0316** | **0.0791** | **0.0912** | **0.0639** | **0.0519** | **0.1089** | **0.0420** |

**Implementation**    For models based on ViT architecture (Ours, DPT, InvPT), we use BEiT-B (Bao et al., 2021) backbone as image encoder, which is pre-trained on ImageNet-22k (Deng et al., 2009) with self-supervision. For the other baselines (DPGNet, HSNet, VAT), as they rely on convolutional encoder and it is nontrivial to transfer them to ViT, we use ResNet-101 (He et al., 2016) backbone pre-trained on ImageNet-1k with image classification, which is their best-performing configuration. During episodic training, we perform task augmentation based on color jittering and MixUp (Zhang et al., 2018) to increase the effective number of training tasks. For all few-shot learning models, we further apply label channel split as described in Section 2. Further details are in Appendix B.

**Evaluation Protocol**    We use the train/val/test split of the Taskonomy-tiny partition provided by Zamir et al. (2018). We train all models on the train split, where DPT and InvPT are trained with full supervision of test tasks $\mathcal{T}_{test}$ and the few-shot models are trained by episodic learning of training tasks $\mathcal{T}_{train}$ only. The final evaluation on test tasks $\mathcal{T}_{test}$ is done on the test split. During the evaluation, all few-shot models are given a support set randomly sampled from the train split, which is also used for task-specific adaptation of VTM as described in Section 3.3. For evaluation on semantic segmentation (SS), we follow the standard binary segmentation protocol in few-shot semantic segmentation (Shaban et al., 2017) and report the mean intersection over union (mIoU) for all classes. For tasks with continuous labels, we use the mean angle error (mErr) for surface normal prediction (SN) (Eigen & Fergus, 2015) and root mean square error (RMSE) for the others.

## 5.2    RESULTS

In Table 1, we report the 10-shot performance of our model and the baselines on ten dense prediction tasks. Our model outperforms all few-shot baselines by a large margin, and is competitive with supervised baselines on many tasks. In Figure 2, we show a qualitative comparison where the few-shot baselines catastrophically underfit to novel tasks while our model successfully learns all tasks. We provide further qualitative comparisons of ours and the baselines in Appendix C.3.

The large performance gap between the few-shot learning baselines and our model can be explained by two factors. **(1)** The core architectural component of HSNet and VAT (feature masking) implicitly relies on the discrete nature of labels, and thus fails to learn tasks with continuous labels whose values are densely distributed. **(2)** Since the baselines are designed to solve tasks without any adaptation of their parameters, the core prediction mechanism (*e.g.*, hypercorrelation of HSNet and VAT, kernel of DGPNet) is fixed and cannot adapt to different semantics of tasks. Unlike the baselines, our model is general for all dense prediction tasks, and has a flexible task adaptation mechanism of the similarity in matching. Our model can also be robustly fine-tuned on few-shot support, thanks to the parameter efficiency of adaptation ($0.28\%$ of all parameters; see Table 7 for comparison with supervised baselines). To demonstrate how VTM performs task adaptation, we visualize the attention of the matching module (Eq. 4). Figure 4 shows that, when adapted for different tasks, the model flexibly changes the similarity to attend to support patches appropriate for the given task.

Surprisingly, with $< 0.004\%$ of the full supervision (10 labeled images), our model even performs better than fully-supervised InvPT on some tasks (*e.g.*, SS and SN). This can be attributed to the robust matching architecture equipped with a flexible adaptation mechanism that enables efficient knowledge transfer across tasks. In ablation study, we show that our model can be further improved by increasing the support size, reaching or sometimes surpassing supervised baselines in many tasks.

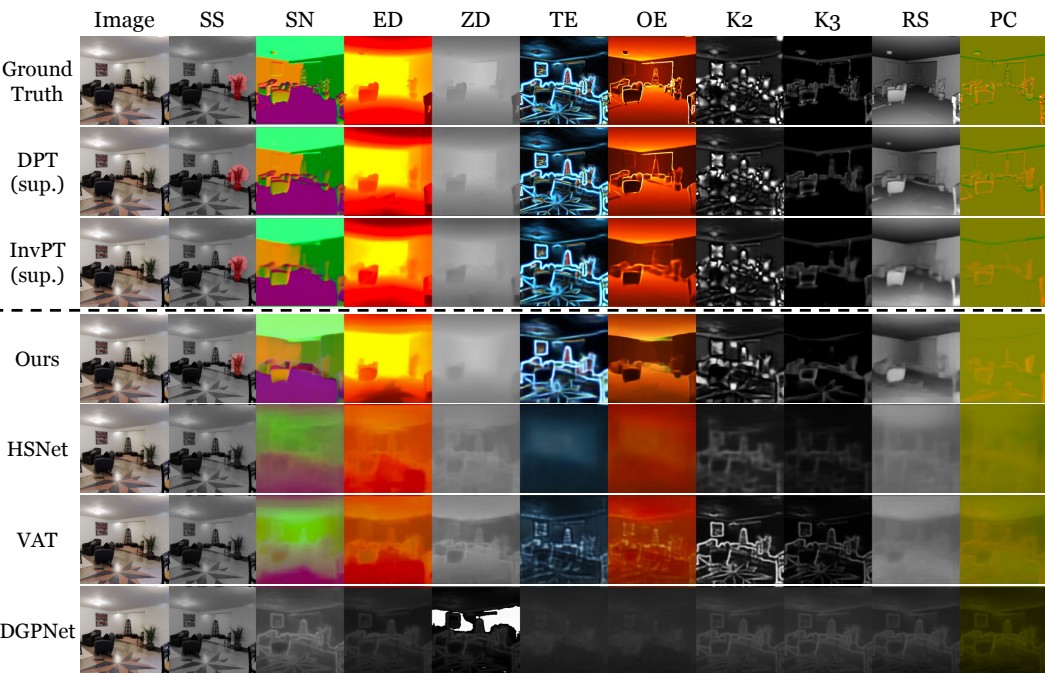

Figure 2: Qualitative comparison of few-shot learning methods in 10-shot evaluation for ten dense prediction tasks in Taskonomy. While other approaches fail, our model successfully few-shot learns all novel tasks with diverse semantics and different label representations.

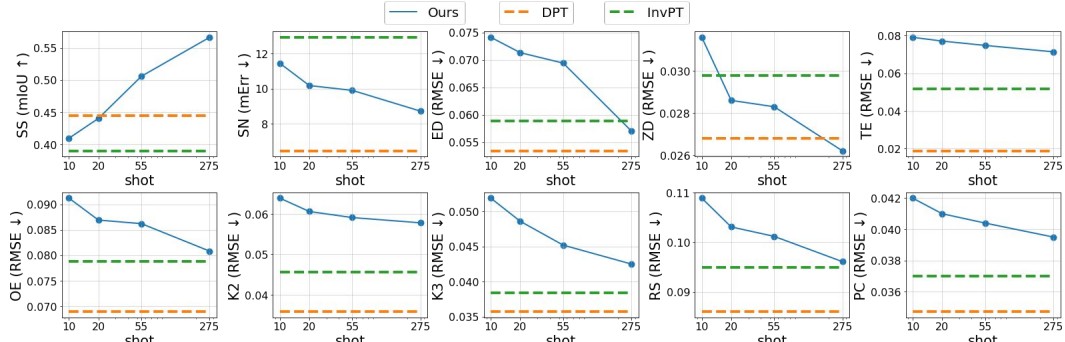

Figure 3: Performance of VTM on various shots. In general, VTM consistently improves performance as more supervision is given, and even surpasses fully supervised baselines on many tasks.

### 5.2.1 ABLATION STUDY

**Component-Wise Analysis** To analyze the effectiveness of our design, we conduct an ablation study with two variants. (1) **Ours w/o Adaptation** does not adapt the similarity for each task in the non-parametric matching. This variant shares all parameters across all training tasks $\mathcal{T}_{\text{train}}$ as well as test tasks $\mathcal{T}_{\text{test}}$, and is evaluated without fine-tuning. (2) **Ours w/o Matching** predicts the query label tokens directly from the query image tokens, replacing the matching module with a parametric linear mapping at each hierarchy. This variant retains the task-specific parameters of ours, thus identical in terms of task adaptation; it utilizes the separate sets of task-specific biases in the image encoder to learn the training tasks $\mathcal{T}_{\text{train}}$, and fine-tune it on the support set of $\mathcal{T}_{\text{test}}$ in the test time.

The results are in Table 2. Both variants show lower performance than ours in general, which demonstrates that incorporating non-parametric matching and task-specific adaptation are both beneficial for universal few-shot dense prediction problems. It seems that the task-specific adaptation is a crucial component for our problem, as Ours w/o Adaptation suffers from the high discrepancy between training and test tasks. For some low-level tasks whose labels are synthetically generated by applying a computational algorithm on an RGB-image (*e.g.*, TE, K2), Ours w/o Matching achieves slightly better performance than Ours. Yet, for tasks requiring a high-level knowledge of object se-

Table 2: Ablation study on matching and task-specific adaptation.

| Model | Tasks | | | | | | | | | |
|---|---|---|---|---|---|---|---|---|---|---|
| | Fold 1 | | Fold 2 | | Fold 3 | | Fold 4 | | Fold 5 | |
| | SS | SN | ED | ZD | TE | OE | K2 | K3 | RS | PC |
| | mIoU ↑ | mErr ↓ | RMSE ↓ | RMSE ↓ | RMSE ↓ | RMSE ↓ | RMSE ↓ | RMSE ↓ | RMSE ↓ | RMSE ↓ |
| Ours w/o Matching | 0.2681 | 13.0704 | 0.1111 | 0.0404 | **0.0778** | 0.1061 | **0.0613** | 0.0537 | 0.1559 | 0.0445 |
| Ours w/o Adaptation | 0.0002 | 23.4212 | 0.1515 | 0.0641 | 0.1513 | 0.1152 | 0.1110 | 0.0625 | 0.2033 | 0.0632 |
| Ours | **0.4097** | **11.4391** | **0.0741** | **0.0316** | 0.0791 | **0.0912** | 0.0639 | **0.0519** | **0.1089** | **0.0420** |

Figure 4: Attention maps of selected level and head for each task given a query patch (red box in the 1st column). Our model flexibly adapts its similarity to attend to appropriate regions for each task (*e.g.*, chairs for semantic segmentation, all planes orthogonal to camera view for surface normal).

mantics (*e.g.*, SS) or 3D space (*e.g.*, ED, OE, K3, RS), the introduction of non-parametric matching fairly improves the few-shot learning performance. Qualitative comparison is in Appendix C.4.

**Impact of Support Size** As our method already performs well with ten labeled examples, a natural question arises: can it reach the performance of fully-supervised approaches if more examples are given? In Figure 3, we plot the performance of VTM by increasing the size of support set from 10 ($< 0.004\%$) to 275 ($0.1\%$). Our model reaches or surpasses the performance of the supervised methods on many tasks with additional data (yet much smaller than full), which implies potential benefits in specialized domains (*e.g.*, medical) where the number of available labels ranges from dozens to hundreds.

**Sensitivity to the Choice of Support Set** As our model few-shot learns a new task from a very small subset of the full dataset, we analyze the sensitivity of the performance to the choice of support set. Table 3 in Appendix reports the 10-shot evaluation results with 4 different support sets disjointly sampled from the whole train split. We see that the standard deviation is marginal. This shows the robustness of our model to the support set, which would be important in practical scenarios.

**Impact of Training Data** The amount and quality of training data is an important factor that can affect the performance of our universal few-shot learner. In Appendix C.1.4, we analyze the effect of the number of training tasks and show that few-shot performance consistently improves when more tasks are added to meta-training data. Also, in a more practical scenario, we may have an *incomplete* dataset where images are not associated with whole training task labels. We investigate this setting in Appendix C.1.5 and show that our model trained with an incomplete dataset still performs well.

## 6 CONCLUSION

In this paper, we proposed and addressed the problem of building a universal few-shot learner for arbitrary dense prediction tasks. Our proposed Visual Token Matching (VTM) addresses the challenge of the problem by extending a task-agnostic architecture of Matching Networks to patch-level embeddings of images and labels, and introducing a flexible adaptation mechanism in the image encoder. We evaluated our VTM in a challenging variant of the Taskonomy dataset consisting of ten dense prediction tasks, and showed that VTM can achieve competitive performance to fully supervised baselines on novel dense prediction tasks with an extremely small amount of labeled examples ($<0.004\%$ of full supervision), and is able to even closer the gap or outperform the supervised methods with fairly small amount of additional data ($\sim 0.1\%$ of full supervision).

**Acknowledgements**   This work was supported in part by the Institute of Information & communications Technology Planning & Evaluation (IITP) (No. 2022-0-00926, 2022-0-00959, and 2019-0-00075) and the National Research Foundation of Korea (NRF) (No. 2021R1C1C1012540 and 2021R1A4A3032834) funded by the Korea government (MSIT), and Microsoft Research Asia Collaborative Research Project.

**Reproducibility Statement**   To help readers reproduce our experiments, we provided detailed descriptions of our architectures in Section B.1, and implementation details of the baseline methods in Table 1 in Section B.2. Since our work proposes the new experiment settings for the few-shot learning of arbitrary dense prediction tasks, we also provide details of the dataset construction process in the main text (Section 5) and the appendix (Section A), which includes details of the data splits, specification of tasks, data preprocessing, and evaluation protocols. We plan to release the source codes and the dataset to ensure the reproducibility of this paper.

**Ethics Statement**   We have read the ICLR Code of Ethics and ensures that this work follows it. All data and pre-trained models used in our experiments are publically available and has no ethical concerns.

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

# APPENDIX

## A  DATASET DETAILS

This section describes the details about the dataset we used in experiments (Section 5).

We use "tiny" version of Taskonomy dataset provided by (Zamir et al., 2018), which consists of images and labels collected from 35 different buildings. We use the train and val split for training and early-stopping, respectively, and use the "muleshoe" building included in the test split for evaluation.

To demonstrate our universal few-shot learner, we use ten dense prediction tasks in Taskonomy dataset (Zamir et al., 2018), which are semantic segmentation (SS), surface normal (SN), Euclidean distance (ED), Z-buffer depth (ZD), texture edge (TE), occlusion edge (OE), 2D keypoints (K2), 3D keypoints (K3), reshading (RS), and principal curvature (PC). All labels are normalized into $[0, 1]$ with task-specific pre-processing. For details on the pre-processing, we refer readers to Zamir et al. (2018). Based on the annotations provided by Taskonomy, we preprocess some tasks to increase the diversity of tasks. Specifically, we modify three single-channel tasks that can be easily augmented: Euclidean distance, texture edge, and occlusion edge.

1. **Texture edge** (TE) labels are generated by applying Sobel edge detector (Kanopoulos et al., 1988) to RGB images, which consists of a Gaussian filter and image gradient computation. The Gaussian filter has two hyper-parameters, namely kernel size and the standard deviation, where adjusting those hyper-parameters yield different *thickness* of detected edges. We use three different sets of hyper-parameters – $(3, 1), (11, 2), (19, 3)$ – to produce 3-channel labels. We give an example of each channel of TE task in Figure 5.

2. **Euclidean distance** (ED) labels consists of pixel-wise depth map, where the depth is computed by the Euclidean distance from each image pixel to the camera's optical center. As this task is very similar to the Z-buffer depth prediction (ZD) whose label pixels are the distance from each image pixel to the camera plane, we augment the ED task by segmenting the depth range and re-normalizing within each segment. Specifically, we compute the 5-quantiles of the pixel-wise depth labels in the whole dataset, then use each quantile as different channels after re-noramlization into $[0, 1]$. Thus the objective of each channel of the augmented ED task is to predict Euclidean distance within a specific range, where the ranges are disjoint for different channels. We give an example of each channel of ED task in Figure 6. To visualize 5-channel labels, we average the first and the second channels as "R"-channel, the third and the fourth channels as "G"-channel, and use the fifth channel as "B"-channel.

3. **Occlusion edge** (OE) labels are similar to texture edge, but they are constructed to depend on only the 3D geometry rather than color or lighting (Zamir et al., 2018). We observe that the channel augmentation by quantiles (that we apply to Euclidean distance task) can fairly diversify the labels. Therefore, we augment the OE labels into 5-channel labels, where we visualize them similar to the ED labels. We give an example of each channel of OE task in Figure 7.

Also, for semantic segmentation, we exclude three classes ("bottle", "toilet", "book"), as little images of the classes are included in the Taskonomy dataset. The 12 classes we used in experiments are: "chair", "couch", "plant", "bed", "dining table", "tv", "mircrowave", "oven", "sink", "fridge", "clock", and "base".

| RGB Image | Kernel size = 3, Std. = 1 | Kernel size = 11, Std. = 2 | Kernel size = 19, Std. = 3 | 3-Channel Visualization |
| --- | --- | --- | --- | --- |

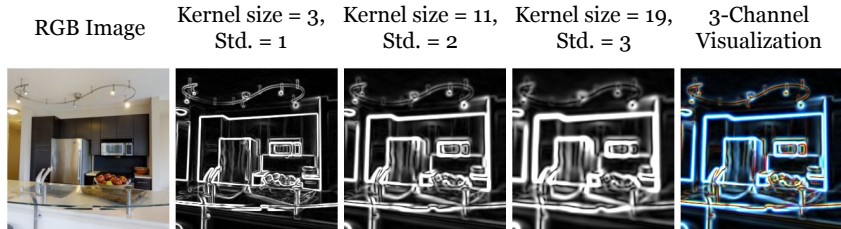

Figure 5: Channel augmentation on texture edge prediction (TE) task. We apply three different sets of hyper-parameters (kernel size, standard deviation) in Sobel edge detector to generate a 3-channel edge task. Second to Fourth columns show the augmented channel with different kernel size and standard deviation, where the last column shows the 3-channel label visualized as RGB.

| RGB Image | 1st 5-quantile | 2nd 5-quantile | 3rd 5-quantile | 4th 5-quantile | 5th 5-quantile | 5-Channel Visualization |
| --- | --- | --- | --- | --- | --- | --- |

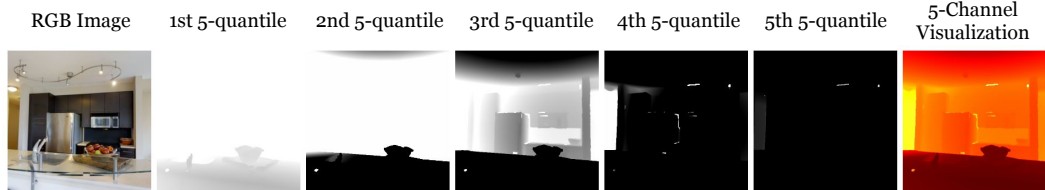

Figure 6: Channel augmentation on Euclidean distance prediction (ED) task. We compute 5-quantiles of the pixel-wise label distribution, and use each $p$-th 5-quantile as each channel after re-normalizing into $[0, 1]$. Second to Fifth columns show the augmented channel with different quantile, where the last column shows the 5-channel label visualized as RGB.

| RGB Image | 1st 5-quantile | 2nd 5-quantile | 3rd 5-quantile | 4th 5-quantile | 5th 5-quantile | 5-Channel Visualization |
| --- | --- | --- | --- | --- | --- | --- |

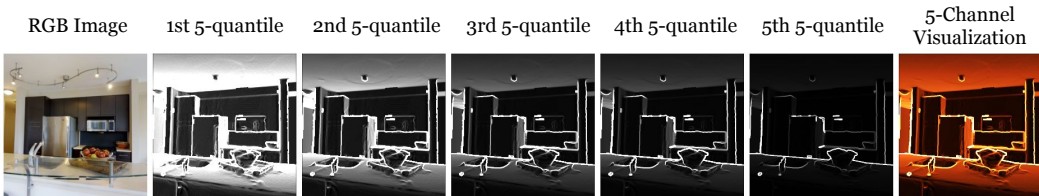

Figure 7: Channel augmentation on occlusion edge prediction (OE) task. We compute 5-quantiles of the pixel-wise label distribution, and use each $p$-th 5-quantile as each channel after re-normalizing into $[0, 1]$. Second to Fifth columns show the augmented channel with different quantile, where the last column shows the 5-channel label visualized as RGB.

## B    IMPLEMENTATION DETAILS

This section describes the implementation details in our experiments (Section 5).

### B.1    ARCHITECTURE DETAILS OF VTM

**Encoders and Decoders**   We employ BEiT-B architecture (Bao et al., 2021) pretrained on Imagenet-22k dataset (Deng et al., 2009) with $224 \times 224$ resolution as our image encoder. For our label encoder and decoder, we follow the DPT-B architecture (Ranftl et al., 2021). Specifically, we use a randomly initialized ViT-B (Dosovitskiy et al., 2020) as label encoder $g$ and extract features from $3, 6, 9, 12$-th layers of the encoder to form multi-level label features (label tokens). Similarly, we extract multi-level image features (image tokens) from $3, 6, 9, 12$-th layers of the image encoder (BEiT). As the DPT-B architecture decodes four-level features using RefineNet-based decoder (Lin et al., 2017), we pass the predicted query label features from matching module at each layer to the decoder. As the label values of tasks in Taskonomy are normalized to $[0, 1]$, we use a sigmoid activation function at the head of the decoder to produce values in $[0, 1]$. To predict semantic segmentation task whose label values are discrete (either $0$ or $1$), we discretize the predicted label with threshold $0.1$.

**Matching Modules**   In the implementation of the matching module with multihead attention, we adopt three conventions in vision transformer (Dosovitskiy et al., 2020) which slightly modifies the equations described in Section 3.2. Recall that the matching module is computed on three input matrices $\mathbf{q} \in \mathbb{R}^{M \times d}$ and $\mathbf{k}, \mathbf{v} \in \mathbb{R}^{NM \times d}$ as follows:

$$\mathrm{MHA}(\mathbf{q}, \mathbf{k}, \mathbf{v}) = \mathrm{Concat}(\mathbf{o}_1, ..., \mathbf{o}_H)w^O, \tag{8}$$

$$\text{where } \mathbf{o}_h = \mathrm{Softmax}\left(\frac{\mathbf{q}w_h^Q(\mathbf{k}w_h^K)^\top}{\sqrt{d_H}}\right)\mathbf{v}w_h^V, \tag{9}$$

where $H$ is number of heads, $d_H$ is head size, and $w_h^Q, w_h^K, w_h^V \in \mathbb{R}^{d \times d_H}, w^O \in \mathbb{R}^{Hd_H \times d}$. First, we perform layer normalization (Ba et al., 2016) before each input projection matrices $w_h^Q, w_h^K, w_h^V$ and after the output projection matrix $w^O$, where we share the layer normalization parameters for $w_h^Q$ and $w_h^K$. Second, we add a residual connection with GELU non-linearity (Hendrycks & Gimpel, 2016) after gathering the outputs from multiple heads as follows:

$$\mathrm{MHA}(\mathbf{q}, \mathbf{k}, \mathbf{v}) = \mathbf{o} + \mathrm{GELU}(\mathbf{o}w^O), \tag{10}$$

$$\text{where } \mathbf{o} = \mathrm{Concat}(\mathbf{o}_1, \mathbf{o}_2, \cdots, \mathbf{o}_H). \tag{11}$$

Finally, we apply Dropout (Srivastava et al., 2014) with rate 0.1 in the attention scores.

### B.2    ARCHITECTURE DETAILS OF BASELINES

**Encoders and Decoders**   For the supervised learning baselines based on transformer encoder (DPT and InvPT), we use the same encoder backbone with ours (BEiT pretrained on ImageNet-22k). We use the decoder of DPT-B configuration in Ranftl et al. (2021) for DPT as ours, and use the original multi-task decoder implementation provided by Ye & Xu (2022) for InvPT. For few-shot learning baselines (HSNet, VAT, DGPNet), we use ResNet-101 (He et al., 2016) pretrained on ImageNet-1k (Deng et al., 2009) as their encoder backbones, which is their best configuration. For the other architectural details, we follow the original implementation of each method provided by Min et al. (2021) (HSNet), Hong et al. (2022) (VAT), and Johnander et al. (2021) (DGPNet).

**Modification on Few-shot Baselines**   As HSNet and VAT are designed for semantic segmentation, we slightly modify their architectures to train them on general dense prediction tasks. Specifically, both models involve a binary masking operation to filter out support image features using their labels (which are assumed to be binary), before computing 4D correlation tensor between support and query feature pixels. For continuous labels of general dense prediction tasks, the binary masking becomes pixel-wise multiplication with labels. However, as the correlation is computed by cosine similarity between feature pixels that is norm-invariant, all non-zero feature pixels with the same

direction are treated in the same manner. This make them unable to discriminate different non-zero label values, *e.g.*, correlation between query and support feature pixels would be the same regardless of the assigned support label values. Therefore, we move the masking operation to after computing the cosine-similarity, so that the models can recognize different non-zero label values through different norms of the masked features by (non-binary) labels.

We use the DGPNet without modification as it is based on a regression method (Gaussian Processes) which is inherently applicable to general dense prediction tasks with continuous labels.

## B.3 TRAINING DETAILS

**Training**   We train all models with 300,000 iterations using the Adam optimizer (Kingma & Ba, 2015), and use *poly* learning rate schedule (Liu et al., 2015) with base learning rates $10^{-5}$ for pre-trained parameters and $10^{-4}$ for parameters trained from scratch. The models are early-stopped based on the validation metric. At each episodic training of iteration, we sample a batch of episodes with size 8. In each episode, we construct a 5-channel task from the training tasks $\mathcal{T}_{\text{train}}$ by first splitting all channels of training tasks and randomly sample 5 channels among them. Then support and query sets are sampled for the selected channels, where we use support and query size of 4 for Ours and DGP, while using 1 for HSNet and VAT as they only supports 1-shot training. To train DPT, we construct a batch of each target task $\mathcal{T}_{\text{test}}$, whose channels are given at once, with batch size 64. To train InvPT, we construct a batch of all ten tasks, whose channels are all given at once, while using batch size 16 due to its large memory consumption.

**Data Augmentation**   We apply random crop (from $256 \times 256$ resolution to $224 \times 224$) and random horizontal flip to images, where the random horizontal flip is applied except for surface normal labels as their values are sensitive to the horizontal direction (flipping images and labels together changes the semantics of the task). As we apply random crop during training, the resolution of test images ($256 \times 256$) differs from the training images. To evaluate the models with consistent resolution, we perform five-crop (cropping the four corners and center of an image) to test query images so that the model also predicts five-cropped labels, then aggregate them by averaging the overlapping regions to produce final prediction for evaluation of resolution ($256 \times 256$). For few-shot models, we apply center crop to support images at test-time.

**Task Augmentation**   For episodic training of few-shot models, we further apply two kinds of task augmentation. First, for each channel of $C$-channel labels sampled at each episode ($C = 5$ in our experiments), we apply random jittering and gaussian blur on each channel independently. Then we apply MixUp (Zhang et al., 2018) on the augmented channels and auxiliary channels which are additionally sampled from the training tasks $\mathcal{T}_{\text{train}}$, to create a linearly interpolated label of two channels. We apply the task augmentation consistently in each episode to preserve the task identity.

## C ADDITIONAL RESULTS

This section provides additional results on our experiments (Section 5).

### C.1 ADDITIONAL RESULTS ON ABLATION STUDY

#### C.1.1 SENSITIVITY TO THE CHOICE OF SUPPORT SET

As discussed in Section 5, we evaluate the 10-shot performance of our VTM with four different support sets that are disjointly sampled from the training data $\mathcal{D}_{\text{train}}$. We report the results in Table 3, which shows that our model is robust to the choice of support set. We use the first support set (#1) in Table 3 for comparison with other baselines or ablated variants in Section 5, due to the huge computational cost for evaluating few-shot baselines HSNet and VAT.

Table 3: Ablation study on the choice of support set. We disjointly sample four different support sets and report the 10-shot performance on each set, with the mean and standard deviation.

| Support set | Tasks | | | | | | | | | |
| | Fold 1 | | Fold 2 | | Fold 3 | | Fold 4 | | Fold 5 | |
| | SS | SN | ED | ZD | TE | OE | K2 | K3 | RS | PC |
| | mIoU ↑ | mErr ↓ | RMSE ↓ | RMSE ↓ | RMSE ↓ | RMSE ↓ | RMSE ↓ | RMSE ↓ | RMSE ↓ | RMSE ↓ |
| #1 | 0.4097 | 11.4391 | 0.0741 | 0.0316 | 0.0791 | 0.0912 | 0.0639 | 0.0519 | 0.1089 | 0.0420 |
| #2 | 0.4190 | 11.8860 | 0.0845 | 0.0338 | 0.0839 | 0.0926 | 0.0629 | 0.0497 | 0.1131 | 0.0437 |
| #3 | 0.3781 | 11.7418 | 0.0776 | 0.0343 | 0.0807 | 0.0944 | 0.0656 | 0.0494 | 0.1101 | 0.0425 |
| #4 | 0.4017 | 11.6203 | 0.0794 | 0.0362 | 0.0799 | 0.0908 | 0.0672 | 0.0502 | 0.1158 | 0.0424 |
| Mean | 0.4021 | 11.6718 | 0.0789 | 0.0340 | 0.0809 | 0.0922 | 0.0649 | 0.0503 | 0.1120 | 0.0427 |
| Std. | 0.0152 | 0.1640 | 0.0038 | 0.0016 | 0.0018 | 0.0014 | 0.0016 | 0.0010 | 0.0027 | 0.0006 |

#### C.1.2 ABLATION STUDY ON TRAINING PROCEDURE

To understand the source of the generalization performance of our method more clearly, we conduct an ablation study on training procedure. We compare four models based on DPT architecture with different training procedures as follows.

- **M1**: Randomly initialized DPT, 10-shot trained.
- **M2**: DPT with BEiT pre-trained encoder, 10-shot fine-tuned.
- **M3** (Ours w/o Matching): DPT with BEiT pre-trained encoder, multi-task trained with task-specific bias tuning, and then 10-shot fine-tuned.
- **M4** (Ours): DPT with BEiT pre-trained encoder, meta-trained with task-specific bias tuning, and then 10-shot fine-tuned.

We summarize the quantitative result in Table 4 and qualitative comparison in Figure 8. First, as expected, we observe that DPT with naive 10-shot training (M1) fails to generalize to the test examples in most of the tasks, except for two 2D texture-related tasks (TE, K2). We conjecture that TE and K2 are "easy" cases in terms of few-shot learning, as they are defined as low-level computational algorithms on RGB images, while other high-level tasks require knowledge about semantics (SS) or 3D space (SN, ED, ZD, OE, K3, RS, PC). Second, we note that BEiT pretraining (M2) largely improves the few-shot generalization performance, allowing the model to produce coarse predictions of the dense labels. However, it still cannot capture object-level fine-grained details in many tasks. Third, we observe that multi-task training and few-shot adaptation, combined with an efficient parameter-sharing strategy of bias tuning (M3, M4), further improves the performance with a clear gap with M2 where the predictions are also qualitatively finer than M2's. Finally, as discussed in Section 5.2.1, M4 still further improves over M3 with a clear gap. This shows that in a few-shot learning setting,

Table 4: 10-shot learning performance of ablated variants of DPT and Ours.

| Model | Tasks | | | | | | | | | |
|---|---|---|---|---|---|---|---|---|---|---|
| | Fold 1 | | Fold 2 | | Fold 3 | | Fold 4 | | Fold 5 | |
| | SS | SN | ED | ZD | TE | OE | K2 | K3 | RS | PC |
| | mIoU ↑ | mErr ↓ | RMSE ↓ | RMSE ↓ | RMSE ↓ | RMSE ↓ | RMSE ↓ | RMSE ↓ | RMSE ↓ | RMSE ↓ |
| M1 | 0.0644 | 21.0976 | 0.1959 | 0.0711 | 0.0995 | 0.1842 | 0.0670 | 0.0600 | 0.2335 | 0.0431 |
| M2 | 0.0582 | 15.8135 | 0.1615 | 0.0530 | 0.1136 | 0.1480 | 0.0948 | 0.0606 | 0.1858 | 0.0431 |
| M3 | 0.2681 | 13.0704 | 0.1111 | 0.0404 | **0.0778** | 0.1061 | **0.0613** | 0.0537 | 0.1559 | 0.0445 |
| M4 | **0.4097** | **11.4391** | **0.0741** | **0.0316** | 0.0791 | **0.0912** | 0.0639 | **0.0519** | **0.1089** | **0.0420** |

Figure 8: Qualitative comparison of Ours and its ablated variants in training procedure. All models use 10 labeled examples for each target task, where M3 and M4 observe additional labeled examples of training tasks (different from the target task) in each fold.

our matching framework and episodic training are more effective than simple multi-task pretraining employed in M3. In summary, we may conclude that the fast generalization of Ours is benefitted from episodic training of various tasks followed by parameter-efficient few-shot adaptation as well as powerful pre-training of the encoder (BEiT).

### C.1.3   FINE-TUNING WITH FULL SUPERVISION

To further explore how our method scales well when a large labeled dataset is given, we also fine-tuned our VTM with full supervision of test tasks. For the fine-tuning, we used the same training dataset as the fully-supervised DPT and employed the episodic fine-tuning objective (Section 3.3). For evaluation, since providing the entire training data as the support set for the matching module is infeasible, we provide a random subset of the training data as the support set to the model. We summarize the result in Figure 9, which extends Figure 3 in Section 5. In most tasks, our model consistently improves when more supervision is given. With full supervision at test tasks, our model performs slightly worse than the DPT baseline in seven tasks and performs better or similarly in the other three tasks. We conjecture that the performance degradation comes from two aspects: (1) the absence of direct input-output connection, *i.e.*, the matching module serves as a bottleneck, and (2) negative transfer from meta-training tasks to test tasks.

### C.1.4   EFFECT OF NUMBER OF TRAINING TASKS

The amount of meta-training tasks is an important factor that can affect the performance of the universal few-shot learner. To verify this, we fixed two test tasks (SS, SN) and trained our VTM on five different subsets of the original eight training tasks (three different subsets with two tasks and two different subsets with five tasks). We summarize the results in the Table 5. As expected, the performance consistently improves as we increase the number of training tasks. We also note that the

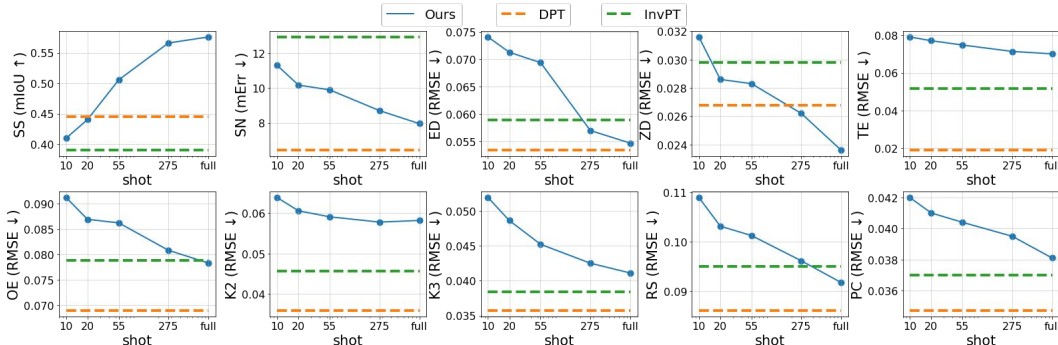

Figure 9: Performance of VTM on various shots. In general, VTM consistently improves performance as more supervision is given, and even surpasses fully supervised baselines on many tasks.

few-shot performance becomes sensitive to the choice of training tasks when their number is small (two), presumably as the model becomes reliant on training tasks more correlated to test tasks, while the variance decreases substantially when more training tasks are added. In addition, the experiment with incomplete training data (Appendix C.1.5) shows the potential ability of our methods in more realistic settings where the training dataset is formed by a combination of different task-specific datasets. From these results, we expect that our model can further enhance its universality on few-shot learning by utilizing a combined training dataset of much more diverse tasks, which we leave as future work.

Table 5: 10-shot learning performance of Ours with various number of training tasks.

| Number of Training Tasks | Tasks | |
| | Fold 1 | |
| | SS | SN |
| | mIoU ↑ | mErr ↓ |
| --- | --- | --- |
| 2 | 0.2878 ± 0.0565 | 17.3947 ± 4.8742 |
| 5 | 0.3919 ± 0.0132 | 12.6769 ± 0.1235 |
| 8 | **0.4097** | **11.4391** |

### C.1.5 EPISODIC TRAINING WITH INCOMPLETE DATASET

It would make our method more practical if the model could learn from an incomplete dataset where images are not associated with whole training task labels. To see how our framework extends to such incomplete settings, we conducted an additional experiment. We simulate the extreme case of incomplete data by partitioning the training images, such that each image is associated with only a single task out of 8 training tasks. Specifically, we partitioned the buildings in Taskonomy into eight groups – each corresponds to a different training task. As this reduces the effective size of training data by the number of training tasks (1/8 in our case), we also train a baseline where we use complete data but use only 1/8 of the training images (for each building, we discard 7/8 of the images). The results are summarized in Table 6. We can see that the performance degradation is marginal when we give incomplete data, which implies that our method can be promising in handling realistic scenarios where the training data is a collection of heterogeneous datasets with different label annotations.

### C.2 FURTHER ANALYSIS

### C.2.1 PARAMETER-EFFICIENCY ANALYSIS

We report the number of task-specific and shared parameters of our VTM and two supervised baselines, DPT and InvPT, to compare how our task adaptation is parameter-efficient. As DPT is a single-task learning model, no parameters are shared across tasks and the whole network should be trained independently for every new task. InvPT, which is a multi-task learning model, shares a large

Table 6: 10-shot learning performance of Ours trained with incomplete and complete multi-task dataset.

| | Tasks | |
| | Fold 1 | |
| Training Data | SS | SN |
| | mIoU ↑ | mErr ↓ |
|---|---|---|
| Incomplete (one task per building) | 0.3559 | 13.6207 |
| Complete (1/8 training data) | 0.3980 | 12.1633 |
| Complete (whole training data) | 0.4097 | 11.4391 |

portion of its parameters across tasks (*e.g.*, encoder backbone), still consumes many parameters for each task in the decoder.

Due to the extensive amount of parameter-sharing, our method is also promising in continual learning setting. As all task-specific knowledge is included in the bias parameters of the image encoder, the knowledge acquired from past tasks can be recalled without forgetting by keeping the corresponding bias parameters and switching to them whenever a past model is needed. We especially note that the size of bias parameters is fairly small (288 KB, which amounts to keeping about 3 labeled images of 256x256 resolution for each task). This allows our model to retain past knowledge very efficiently by keeping the tuned bias parameters plus a few-shot support set, whose external memory requirement is far less compared to memory-based approaches in continual learning that keep hundreds of images (Bang et al., 2021; Wang et al., 2022). While the continual learning setting is not our main focus, applying our method to a continual learning setting would be an interesting future direction.

Table 7: Number of task-specific and shared parameters for a single-channel task (in million).

| Model | Task-Specific | Shared |
|---|---|---|
| DPT (supervised learning) | 110.55 | 0 |
| InvPT (multi-task learning) | 24.57 | 106.75 |
| Ours (few-shot learning) | 0.0703 | 202.95 |

### C.2.2 COMPUTATION COST ANALYSIS

To analyze how our method is computationally efficient compared to supervised DPT, we measured the MACs (multiply–accumulate operations) of our model and DPT using an open-source python library thop [3]. We report the results in Table 8. Having encoded the support set (e.g., 10-shot), we can see that the computational cost of our model's inference on a single query image is about 30% larger than the cost of DPT's, due to the Matching part.

Table 8: MACs of Ours and DPT on a single-query inference for a single-channel task.

| Model | MACs (G) |
|---|---|
| DPT | 30.15 |
| Ours after encoding support (10-shot) | 38.79 |

---

[3]https://github.com/Lyken17/pytorch-OpCounter

### C.2.3 ROLE OF ATTENTION HEADS

To analyze the role of attention heads, in Figure 10, we visualized the attention maps for each head over support images for a given query patch, feature level (3rd level in this example), and task (RS in this example). The figure shows that each head attends to different regions of the support images. Moreover, we can find some patterns in heads; for example, the first head tends to attend to flat areas of the scene, such as the floor or ceiling (low-frequency features), while the third head tends to attend to objects, such as couch or plant (high-frequency features). To further verify the benefit of multi-head attention in the matching module, we also trained our VTM with single head in the matching modules. The result is summarized in the table below and Table 9. We can see the performance drop in both SS and SN tasks, which supports that exploiting multiple heads benefits our matching framework.

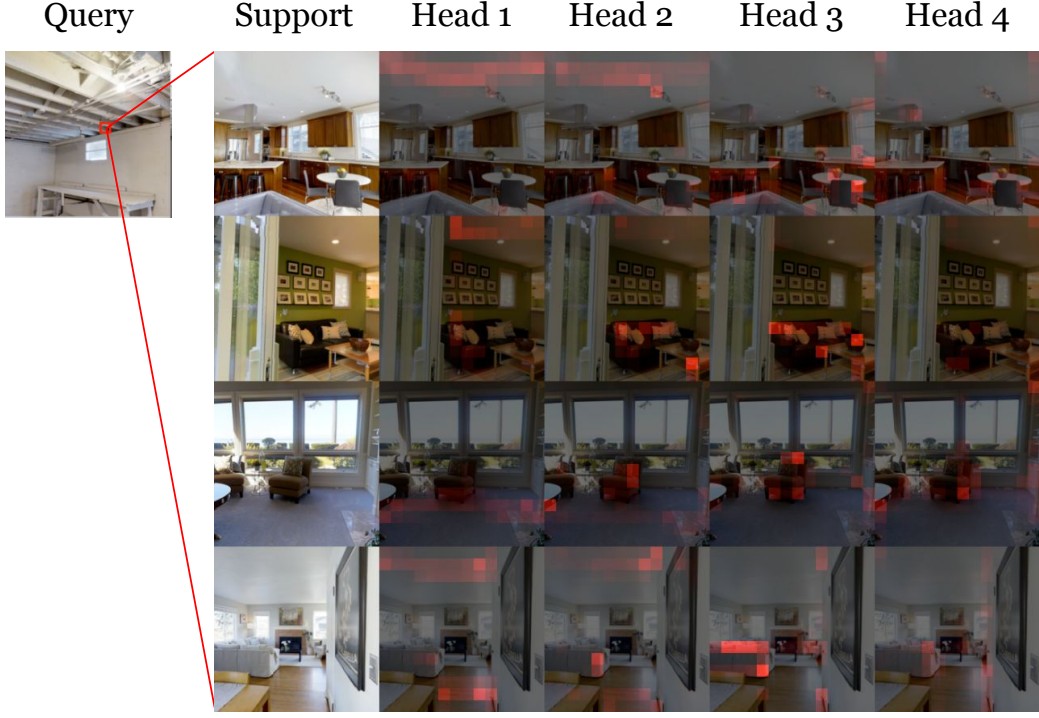

Figure 10: Visualization of multi-head attention maps of VTM. Here we visualize the matching module at 3rd level for reshading (RS) task.

Table 9: 10-shot learning performance of Ours with different number of attention heads in Matching module.

| Number of Attention Heads | Tasks | |
|---|---|---|
| | Fold 1 | |
| | SS | SN |
| | mIoU ↑ | mErr ↓ |
| 1 | 0.3702 | 12.5936 |
| 4 | **0.4097** | **11.4391** |

## C.3 ADDITIONAL QUALITATIVE COMPARISON WITH BASELINES

We provide additional results on the qualitative evaluation of our model and the baselines. Figure 11-14 show visualizations on different query image and support set, where we vary the class of semantic segmentation task included in each support. The result shows consistent trends of that we discussed in Section 5. Ours is competitive to the fully supervised baselines (DPT and InvPT), while the other few-shot baselines (HSNet, VAT, DGPNet) fail to learn different dense prediction tasks.

In Figure 12, even the GT label for semantic segmentation ("couch" class) is noisy as it is a pseudo-label generated by a pre-trained segmentation model (Zamir et al., 2018), our model successfully segments two couches present in the figure. This can be attributed to the task-agnostic architecture of VTM based on non-parametric matching.

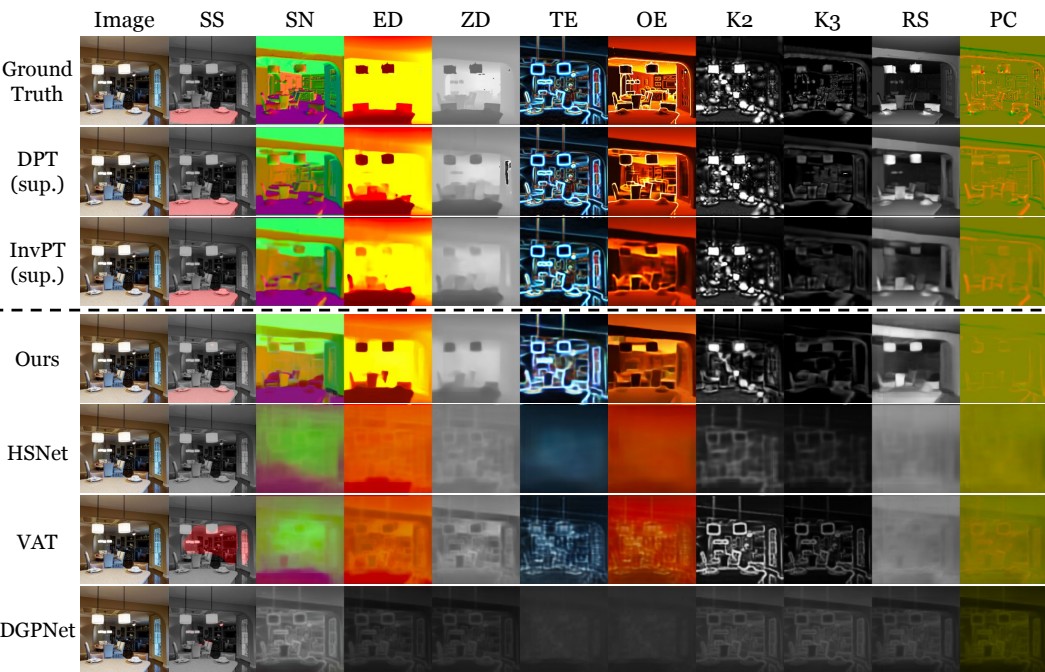

Figure 11: Additional results of qualitative comparison between Ours and the baselines.

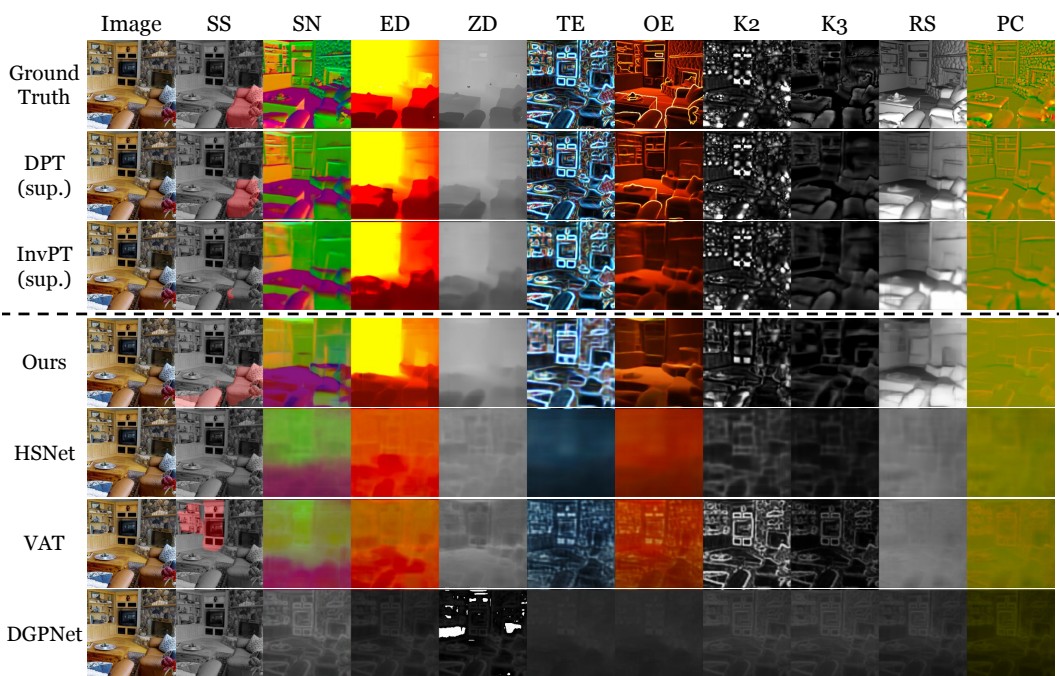

Figure 12: Additional results of qualitative comparison between Ours and the baselines.

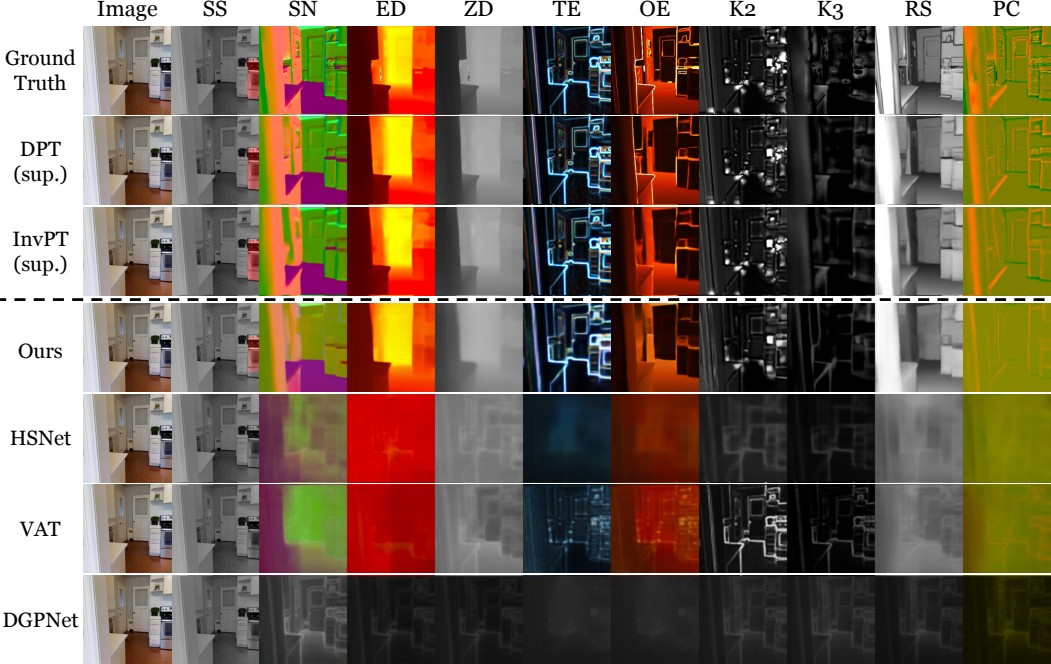

Figure 13: Additional results of qualitative comparison between Ours and the baselines.

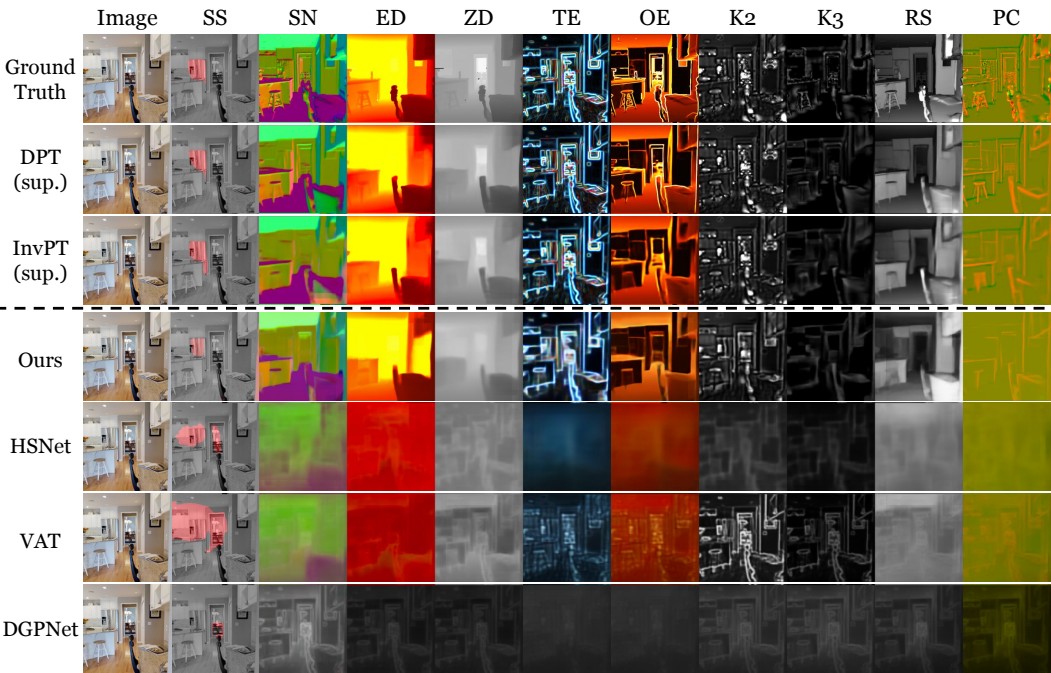

Figure 14: Additional results of qualitative comparison between Ours and the baselines.

## C.4 ADDITIONAL QUALITATIVE COMPARISON WITH OUR VARIANTS

We also provide additional results on the qualitative evaluation of our model and our ablated variants, Ours w/o Matching and Ours w/o Adaptation. Figure 15-18 show visualizations on different query image and support set. The results show a consistent trend with the quantitative results in Table 1. Interestingly, our method without adaptation already exhibits some degree of adaptation to the unseen tasks even without fine-tuning and task-specific components, showing that the non-parametric architecture of our model and the parameter sharing derived from is appropriate to learn generalizable knowledge to understand the novel tasks. On the other hand, adding a task-specific component and adaptation mechanism to the model allows more dramatic improvement in understanding novel tasks from few-shot examples, showing the importance of the adaptation mechanism in our task. Finally, we observe that equipping the matching mechanism with the adaptation module provides much sharper and fast adaptation to the unseen tasks, which verifies our claims.

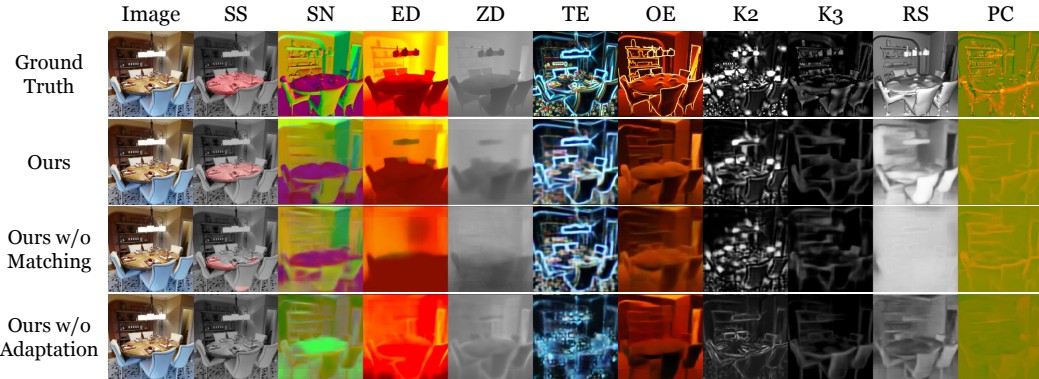

Figure 15: Additional results of qualitative comparison between Ours and its ablated variants.

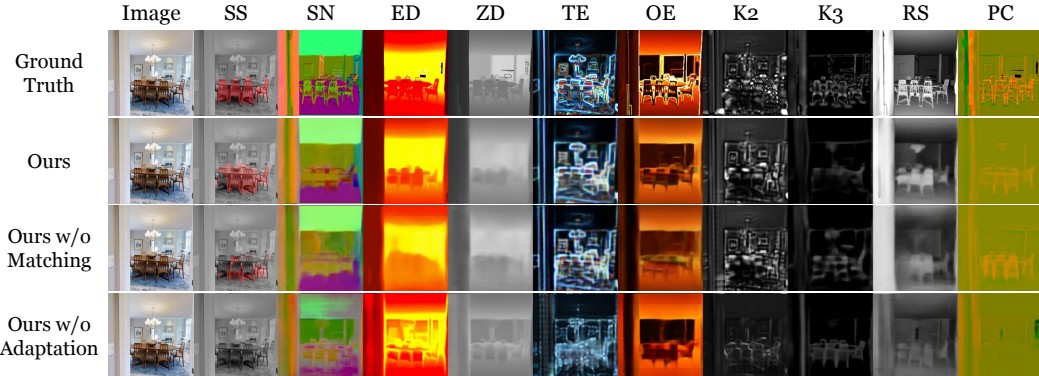

Figure 16: Additional results of qualitative comparison between Ours and its ablated variants.

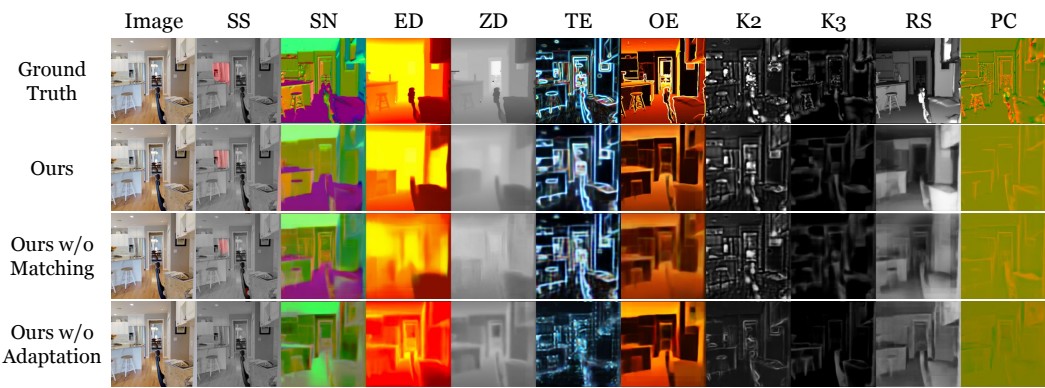

Figure 17: Additional results of qualitative comparison between Ours and its ablated variants.

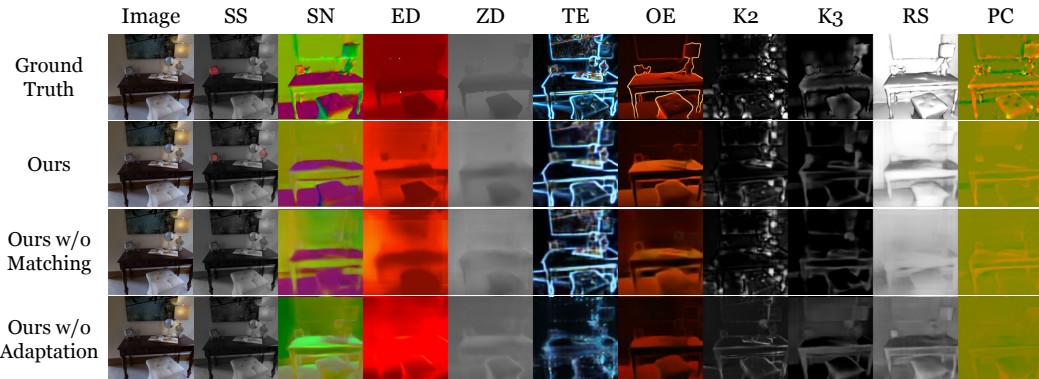

Figure 18: Additional results of qualitative comparison between Ours and its ablated variants.

