# OpenReview forum: "Universal Few-shot Learning of Dense Prediction Tasks with Visual Token Matching"
_ICLR.cc/2023/Conference — ICLR 2023 notable top 5%_

### Official Review · Reviewer_arpe · 2022-10-25

**Confidence:** 3
**Clarity, Quality, Novelty And Reproducibility:** See above.
**Correctness:** 3
**Technical Novelty And Significance:** 4
**Empirical Novelty And Significance:** 4
**Recommendation:** 10

**Strength And Weaknesses:**

This paper was clear, presented an interesting new problem and has excellent results.  I much enjoyed reading it.  Questions and comments I had are below:

* What is the computational cost of the system, compared to a fully-supervised model (DPT) inference?  Given already-encoded support set features, what the cost to execute one query inference, including query image encoding and matching?

* Was any loss or gradient balancing needed between task types (between the different loss measures), and how was this applied?

* While the motivation of nearest-neighbor lookup is clear, I think the multiheaded attention is more sophisticated than simple NN on key to label tokens, particularly with multi-scale features, as it is here.  Since each head outputs a projection of values, which are linearly combined, based on a projection of keys and queries, different parts of different support images can be used for different components of the label token.  In particular, there can be differences in scale:  Low frequency parts of the token requiring wide contexts can match to one relevant set of support features, while high frequency edge transitions requiring local context *for the same query prediction and task* might match to other support features, even from support images having completely different global context or layout.  How much have you seen this occur, and how important is the number of heads in the MHA matching?  What other sorts of components have this behavior?  Some discussion of this would be nice.

* intro sec 1:  "the model learns the similarity in image patches that capture the similarity in label patches":  This seems it would require a relatively large number of tasks of different types, in order to cover differently relevant ways of matching, and indeed the 80/20 5-fold cross-eval does seem to supply a fair number.  What about varying the number training task types --- in what ways does the method fail?

* sec 5.2: discussion on why the method outperforms few-shot semantic segmentation baselines:  the two points here, while likely large factors for most tasks, don't explain the performance difference for semantic segmentation task, where the model described in the paper also performs better.  Do you have any ideas about this difference, and can it explain a large portion of the gains for other tasks as well?

* Decomposing the problem to single-channel output tasks also may have a factorization effect, that stands to improve performance --- does it?

* Are biases in the encoder tuned for each output channel separately, or jointly for all output channels in the test task (for multi-channel tasks like SN)



**Summary Of The Paper:**

This paper presents a method for few-shot learning on wide varieties of novel of dense prediction tasks (not just semantic classes) on images.  The method applies meta-learning episodes to a vision transformer, trained to predict label tokens, which are decoded to final labels using a convolutional net.  The final attention step of the transformer is applied using features from support images as the keys, but encoded features ("label tokens") as the values; this effectively transfers labels from the support set to the query image, as combinations of support label tokens, which are subsequently decoded to final labels.  The system is evaluated using 10 tasks from the Taskonomy dataset, comparing against fully-supervised baselines as well as current few-shot semantic segmentation methods, finding excellent performance in both 10-shot and larger 200-shot settings, sometimes competitive with fully-supervised training in only the 10-shot setting.


**Summary Of The Review:**

This is an excellent paper that describes a very effective approach to a new and interesting problem.

---

> ### Author Response · Authors · 2022-11-18
> **Rebuttal from Authors of Paper5139 to Reviewer arpe (2/2)**
>
> ---
>
> > **Q4.** intro sec 1: "the model learns the similarity in image patches that capture the similarity in label patches": This seems it would require a relatively large number of tasks of different types, in order to cover differently relevant ways of matching, and indeed the 80/20 5-fold cross-eval does seem to supply a fair number. What about varying the number training task types --- in what ways does the method fail?
>
> **A4.** We appreciate the reviewer’s comment and added the following discussion to the paper. Indeed, the amount of meta-training tasks is an important factor that can affect the performance of the universal few-shot learner. To verify this, following the reviewer’s suggestion, we fixed two test tasks (SS, SN) and trained our VTM with 2, 5, and 8 training tasks. To account for the potential variance that arises from the choice of training tasks, we choose multiple subsets of the training tasks of sizes 2 and 5, then report the mean and standard deviation. We summarize the results in the table below and Table 5 in Appendix C.1.4.
>
> |  Number of Training Tasks | SS (mIoU ↑) | SN (mErr ↓) |
> |:---:|:---:|:---:|
> | 2 | 0.3081 ± 0.0932 | 16.1029 ± 4.7710 |
> | 5 | 0.3702 ± 0.0029 | 12.7073 ± 0.2592 |
> | 8 | **0.4210** | **12.4863** |
>
> As expected, the performance consistently improves as we increase the number of training tasks. We also note that the few-shot performance becomes sensitive to the choice of training tasks when their number is small (two), presumably as the model becomes reliant on training tasks more correlated to test tasks, while the variance decreases substantially when more training tasks are added.
>
> In addition, we kindly refer the reviewer to our additional experiment with incomplete training data (Appendix C.1.5), which shows the potential ability of our methods in more realistic settings where the training dataset is formed by a combination of different task-specific datasets. From these results, we expect that our model can further enhance its universality on few-shot learning by utilizing a combined training dataset of much more diverse tasks, which we leave as future work.
>
> &nbsp;
>
> ---
>
> > **Q5.** sec 5.2: discussion on why the method outperforms few-shot semantic segmentation baselines: the two points here, while likely large factors for most tasks, don't explain the performance difference for semantic segmentation task, where the model described in the paper also performs better. Do you have any ideas about this difference, and can it explain a large portion of the gains for other tasks as well?
>
> **A5.** Presumably, this is because the few-shot semantic segmentation baselines are meta-trained on eight *continuous* tasks and then meta-tested on semantic segmentation. This is a totally different setting from conventional few-shot segmentation, where meta-training involves semantic segmentation tasks only. As few-shot segmentation methods are mainly designed specifically for semantic segmentation, they fail to meta-learn from various continuous tasks, resulting in poor generalization to few-shot semantic segmentation at test time.
>
> &nbsp;
>
> ---
>
> > **Q6.** Decomposing the problem to single-channel output tasks also may have a factorization effect, that stands to improve performance --- does it?
>
> **A6.** To the best of our understanding, the factorization effect of channel decomposition mentioned by the reviewer stands for decomposing a complex multi-channel task into simpler single-channel tasks, thus making the problem easier. As suggested by the reviewer, we conjecture that the factorization effect can improve performance in our few-shot learning setting, since an adaptation to a multi-channel task would require more labeled data to generalize well than an adaptation to a single-channel task due to the higher dimensionality. Furthermore, as we tune bias parameters for each channel separately, learning the optimal bias parameters of each channel might be easier than learning those of all channels jointly. We hope this answers the reviewer’s question and are happy to elaborate more if the reviewer wants any further discussions.
>
> &nbsp;
>
> ---
>
> > **Q7.**
> Are biases in the encoder tuned for each output channel separately, or jointly for all output channels in the test task (for multi-channel tasks like SN)
>
> **A7.** As described in Section 2, we treat each multi-channel task as multiple single-channel tasks. Thus the bias parameters for each output channel are tuned separately (*e.g.*, tune three separate sets of bias parameters for each channel of SN task).

---

> > ### Comment · Reviewer_arpe · 2022-12-03
> > **responses**
> >
> > Thanks for the detailed responses.  They address all my questions well.  The new experiments on number of training tasks and number of attention heads are particularly interesting to see.

---

> ### Author Response · Authors · 2022-11-18
> **Rebuttal from Authors of Paper5139 to Reviewer arpe (1/2)**
>
> ---
>
> > **Q1.** What is the computational cost of the system, compared to a fully-supervised model (DPT) inference? Given already-encoded support set features, what the cost to execute one query inference, including query image encoding and matching?
>
> **A1.** We measured the total number of multiply–accumulate operations (MACs) of our model and DPT using an open-source python library thop (https://github.com/Lyken17/pytorch-OpCounter). We report the results in the table below and Table 8 in Appendix C.2.2.
>
> | Model | MACs (G) |
> |---|---|
> | Ours after encoding support (10-shot) | 38.79 |
> | DPT | 30.15 |
>
> Having encoded the support set (*e.g.*, 10-shot), we can see that the computational cost of our model’s inference on a single query image is about 30% larger than the cost of DPT’s. The increased cost is introduced by the Matching process that performs multi-head attention on the query and support tokens, which does not dominate the overall cost of the inference.
>
> &nbsp;
>
> ---
>
> > **Q2.** Was any loss or gradient balancing needed between task types (between the different loss measures), and how was this applied?
>
> **A2.** We did not use any loss or gradient balancing between task types. Losses for all tasks are given equally, including both continuous (L1 loss) and discrete (cross-entropy loss) ones. In all our experiments, the episodic training was stable without any loss or gradient balancing.
>
> &nbsp;
>
> ---
>
> > **Q3.** While the motivation of nearest-neighbor lookup is clear, I think the multiheaded attention is more sophisticated than simple NN on key to label tokens, particularly with multi-scale features, as it is here. Since each head outputs a projection of values, which are linearly combined, based on a projection of keys and queries, different parts of different support images can be used for different components of the label token. In particular, there can be differences in scale: Low frequency parts of the token requiring wide contexts can match to one relevant set of support features, while high frequency edge transitions requiring local context for the same query prediction and task might match to other support features, even from support images having completely different global context or layout. How much have you seen this occur, and how important is the number of heads in the MHA matching? What other sorts of components have this behavior? Some discussion of this would be nice.
>
> **A3.** We appreciate the valuable comment and added the following discussion to the paper. To analyze the role of attention heads, in Figure 10 in Appendix C.2.3, we visualized the attention maps for each head over support images for a given query patch, which shows that each head attends to different regions of the support images. Moreover, we can find some patterns in heads; for example, one head tends to attend to flat areas of the scene, such as the floor or ceiling (low-frequency features), while another head tends to attend to objects, such as a couch or plant (high-frequency features).
>
> To further verify the benefit of multi-head attention in the matching module, we also trained our VTM with single head in the matching modules. The result is summarized in the table below and Table 9 in Appendix C.2.3.
>
> | Number of Attention Heads | SS (mIoU ↑) | SN (mErr ↓) |
> |:---:|:---:|:---:|
> | 1 | 0.3264 | 13.2745 |
> | 4 | 0.4210 | 12.4863 |
>
> We can see that the performance drops in both SS and SN tasks, which supports that exploiting multiple heads benefits our matching framework.
>
> &nbsp;

---

### Official Review · Reviewer_hDLX · 2022-10-28

**Confidence:** 4
**Correctness:** 4
**Technical Novelty And Significance:** 4
**Empirical Novelty And Significance:** 4
**Recommendation:** 8

**Clarity, Quality, Novelty And Reproducibility:**

The paper is well-written and clear to understand. It is an original and novel work to the best of my knowledge. I'm not sure about the reproducibility and I encourage the authors to share the code including experiment configs upon acceptance.

**Strength And Weaknesses:**

Strengths:
- The idea is novel and intuitive. I very much enjoy many designs of the architecture. Just to list a few:
  - The image encoder has shared and separate portion of parameters across different tasks. Leveraging the bias-tuning idea, the few-shot learning optimizing biases should be very efficient.
  - Multi-channel labels are split into single-channel ones. This design enables flexible joint learning of multiple tasks
  - The token matching module provides a straightforward way of aggregating labels from the support set. It is also multi-resolutional to support learned aggregation on different abstraction levels.
- The results are impressive. VTM outperforms few-shot learning baselines significantly. The qualitative results also appear to be better than baselines.
- The paper is well-written and designs are well-motivated.

Weaknesses:
- Limitation on model architecture. The method requires a same architecture for all tasks. This design may lead to suboptimal architecture for different tasks. I don't see an easy way to enable flexibility of architecture for different tasks within the proposed paradigm.
- Limitation on training data. I could be wrong on my understanding of the experimental setting but apparently in the meta training, labels of 8 training tasks are all used for each image example. Although Taskonomy supports such a setting with comprehensive labels, in the real world it could be too expensive to obtain so many dense labels for each image. Have you tried an alternative setting, where in the meta training stage, each training image only comes with one type of annotation? This setting will be more realistic and can be extended to real-world datasets by combining multiple datasets for different dense understanding tasks.

**Summary Of The Paper:**

The paper proposes a few-shot multitask learner, Visual Token Matching (VTM), for dense prediction tasks. VTM is composed of four components including an image encoder which contains shared and separate parameters for different tasks, a label encoder which is kept the same across tasks, a matching module and a label decoder to read out desired dense predictions. The method is benchmarked on Taskonomy dataset and shows significant improvement over prior work of few-shot multitask learning.

**Summary Of The Review:**

Overall this is an inspiring paper with a well-designed and effective method for few-shot multitask learning on dense prediction tasks. Despite aforementioned limitations of the architecture and the training data, it is a work worth reading by a large community. I hope the authors can provide more thoughts regarding the weaknesses during the rebuttal phase.

---

> ### Author Response · Authors · 2022-11-18
> **Rebuttal from Authors of Paper5139 to Reviewer hDLX**
>
> ---
>
> > **Q1.** Limitation on model architecture. The method requires a same architecture for all tasks. This design may lead to suboptimal architecture for different tasks. I don't see an easy way to enable flexibility of architecture for different tasks within the proposed paradigm.
>
> **A1.** We agree with the reviewer that task-specific inductive bias offered by tailored architecture can benefit specific tasks. But by the no-free-lunch theorem, there would be a necessary trade-off between architectural optimality and universality in few-shot learning. Indeed, one major benefit of task-agnostic architecture (as in our work) is that we do not have to design different architectures and use a single, universal architecture for any unseen task. Nevertheless, we tried our best to diversify and adapt the matching paths within our architecture task-specifically, in terms of both vertical feature depths (multi-level matching) and horizontal matching paths (multi-head attention) so that each task could exploit the best fitting paths. Enabling architectural flexibility for different tasks would be an interesting future research direction.
>
> &nbsp;
>
> ---
>
> > **Q2.** Limitation on training data. I could be wrong on my understanding of the experimental setting but apparently in the meta training, labels of 8 training tasks are all used for each image example. Although Taskonomy supports such a setting with comprehensive labels, in the real world it could be too expensive to obtain so many dense labels for each image. Have you tried an alternative setting, where in the meta training stage, each training image only comes with one type of annotation? This setting will be more realistic and can be extended to real-world datasets by combining multiple datasets for different dense understanding tasks.
>
> **A2.** We appreciate the valuable comment and added the following discussion to the paper. As pointed out by the reviewer, it would make our method more practical if the model could learn from an incomplete dataset where images are not associated with whole training task labels. To see how our framework extends to such incomplete settings, we conducted an additional experiment. As suggested by the reviewer, we simulate the extreme case of incomplete data by partitioning the training images, such that each image is associated with only a single task out of 8 training tasks. Specifically, we partitioned the buildings in Taskonomy into 8 groups – each corresponds to a different training task. As this reduces the effective size of training data by the number of training tasks (1/8 in our case), we also train a baseline where we use complete data but use only 1/8 of the training images (for each building, we discard 7/8 of the images). The results are summarized in the table below and Table 6 in Appendix C.1.5.
>
> | Training Data | SS (mIoU ↑) | SN (mErr ↓) |
> |:---:|:---:|:---:|
> | Incomplete (one task per building) | 0.3332 | 13.9443 |
> | Complete (1/8 training data) | 0.3549 | 12.6526 |
> | Complete (whole training data) | 0.4210 | 12.4863 |
>
> We can see that the performance degradation is marginal when we give incomplete data, which implies that our method can be promising in handling realistic scenarios where the training data is a collection of heterogeneous datasets with different label annotations.
>
> &nbsp;

---

### Official Review · Reviewer_1qaw · 2022-11-03

**Confidence:** 4
**Correctness:** 4
**Technical Novelty And Significance:** 3
**Empirical Novelty And Significance:** 4
**Recommendation:** 10

**Clarity, Quality, Novelty And Reproducibility:**

The paper is well-written and clear.

The contributions are original and few-shot adaptation to general dense prediction tasks is indeed unexplored (although task-specific few-shot methods exist, e.g. for semantic segmentation).
The proposed method needs to be finetuned for a novel task. I would like to see some discussion in the related work section about methods that do not need to finetune, e.g. some prompting approaches in language (e.g. GPT-3) and vision (e.g. the recent work "Visual Prompting via Image Inpainting" or other similar).
On a technical level, the architecture combines existing modules in an elegant way.

An experienced reader would probably be able to reproduce parts of this work, as it is based on well-known components. If the authors release their code it would be useful to the community.

**Strength And Weaknesses:**

Strengths:

(1) The problem of few-shot multi-tasking for dense prediction is important and unexplored.

(2) The proposed approach is simple and intuitive, without task-specific inductive biases or extreme computational complexity. The experiments show that it works well.

(3) The writing is clear and helps the understanding of the motivation, desiderata, solution.


Weaknesses:

There are some discussions/comparisons that are missing.

(1) Could you please explain the technical differences between the w/o Matching baseline and DPT? Equivalently, if we add a support set on DPT, how different is this from the proposed model?

(2) The proposed architecture is not specifically designed for few-shot settings. In fact, if we compare the w/o Matching variant from Table 2 to the results of the baselines in Table 1, it does pretty well. This is good but also raises the question of how much the fast generalization is a result of the powerful architecture vs the multi-tasking meta-training. These additional experiments would shed light on this:
(a) What is the performance of the proposed model if trained fully-supervised? Does it scale well when a lot of data is there? How does it compare to DPT? (b) How does out-of-the-box DPT work after few-shot finetuning?

(3) From Table 2 it appears that fine-tuning is very important. This is fine as long as we care about the performance on the novel tasks only. But once finetuned, the model could forget the training tasks. This makes the approach less significant from the point of continual learning. Could you evaluate the forgetting, i.e. what is the performance on the training tasks after finetuning on the novel tasks?

(4) Training for more tasks jointly is a harder job for the model but makes the dependency on the support set stronger. How important is that aspect for the proposed model? If you train on 5 tasks and test on 5 vs 8-2 vs 2-8, how does the performance change? It would be nice if you keep 2 novel tasks as fixed and then train the model on 2, 5 and 8 tasks and show how the performance on the two held-out tasks changes.

Questions:

(a) Is the whole support set used in a forward pass? That's fine in 10-shot, but when you have 275 support images, do you still feed all of them at the same time? Would it make sense to finetune on 275 support images but use only a fixed set of them during inference? In that case this approach could scale to large datasets as well.

(b) Why do you think that you need to finetune the model? Isn't the support set enough to provide the task description? Could somehow the support set predict the task-specific parameters to use?

**Summary Of The Paper:**

The paper tackles few-shot dense prediction from image, i.e. tasks where the input is an image and the output is a per-pixel prediction. The authors propose a Transformer-based meta-learning approach: learning some dense prediction tasks in an episodic fashion helps the model generalize to novel tasks. Most of the parameters on the model are shared, so a small-scale finetuning is needed to learn a new task.

The architecture is simple: the target image and the support images are split into patches that are featurized using a Transformer. Then the target patches attend to the support patches. Based on their similarity, a target patch inherits the weighted label encoding of each support patch. The label encodings are summed over all patches into a single vector. Another network upsamples this vector to the full image resolution.
There are two important details here. First, all weights are shared across tasks, except for the bias terms in the image encoder. This forces us to finetune for a new task, but the actual number of parameters is small. Second, supposing that the output has C channels, the proposed model predicts a dense map for each channel separately. This allows generalization to an arbitrary number of channels at test time.

The authors test their approach on Taskonomy, by splitting 10 dense prediction tasks into 8 for training and 2 for testing (5-fold validation). They compare against fully-supervised and few-shot baselines and significantly outperform the few-shot ones while sometimes performing on par with fully-supervised approaches. Their ablation indicates that finetuning for new tasks is necessary and that most of the times their network benefits from context (attending to the support set) to improve performance.

**Summary Of The Review:**

In general I like the motivation and technical aspects of this paper. I feel the problem is important and the approach seems to make a step towards the right directions. However, I feel that some comparisons that could make the arguments stronger are missing. In addition, I am concerned about depending on finetuning, although I recognize that continual learning is not the focus of this work. I'm looking forward to the discussion with the authors and will happily vote for accepting the paper if they address my questions.

After the rebuttal, all my concerns were addressed and I strongly recommend this paper for acceptance.

---

> ### Author Response · Authors · 2022-11-18
> **Rebuttal from Authors of Paper5139 to Reviewer 1qaw (4/4)**
>
> ---
>
> > **Q5.** Is the whole support set used in a forward pass? That's fine in 10-shot, but when you have 275 support images, do you still feed all of them at the same time? Would it make sense to finetune on 275 support images but use only a fixed set of them during inference? In that case this approach could scale to large datasets as well.
>
> **A5.** As pointed out by the reviewer, we cannot increase the support set size for each forward pass indefinitely since the matching module (Figure 1) scales linearly to the number of support images for each given query image. Thus, for our 275-shot experiment, we fine-tune on all 275 support images by minibatching (*e.g.*, sample a minibatch of 100 images and divide it into 25 sub-support and 75 sub-query images) and use a random subset of 100 support images during the final inference (exactly as the mentioned by the reviewer). The same protocol is used to obtain the fine-tuning results with full supervision in the answer above (**A2**). We hope this clarifies the reviewer’s concern and will revise the text to clarify the many-shot fine-tuning and evaluation protocol.
>
> &nbsp;
>
> ---
>
> > **Q6.** Why do you think that you need to finetune the model? Isn't the support set enough to provide the task description? Could somehow the support set predict the task-specific parameters to use?
>
> **A6.** Compared to the tasks addressed by usual few-shot learning methods (*e.g.*, Matching Networks for image-level classification), general dense prediction tasks in computer vision are much more heterogeneous (diverse) in semantics, as described in Section 2.1. Therefore, we expect and design our universal few-shot learner to have more capability of task-specific adaptation compared to traditional models (*e.g.*, Matching Networks or few-shot segmentation baselines in Section 5), for which fine-tuning task-specific parameters is one of the straightforward and effective ways. Indeed, in our VTM framework, fine-tuning enables our model to efficiently and effectively adapt its token similarity function to an arbitrary target task.
>
> As suggested by the reviewer, it is also possible to introduce a hyper-network that predicts the task-specific parameters [3, 4] rather than fine-tuning the parameters from scratch as in our work. However, training such hyper-network to generalize well to unseen dense prediction tasks would be challenging, as our universal few-shot learning consider a much broader range of dense prediction tasks compared to conventional few-shot learning approaches. In this work, we only focus on Bias Tuning, as it is known as an efficient and effective way to tune a pre-trained Transformer [5]. We also note that it can also be seen as an example of the encoder-free method in generative modeling, where latent variables are inferred by gradient descent instead of a neural network [6]. Further exploring how to adapt the matching module task-specifically, *e.g.*, using a hyper-network, would be an interesting future direction.
>
> [3] Requeima, James, et al. "Fast and flexible multi-task classification using conditional neural adaptive processes." Advances in Neural Information Processing Systems 32 (2019).
>
> [4] Wang, Yu-Xiong, Deva Ramanan, and Martial Hebert. "Meta-learning to detect rare objects." Proceedings of the IEEE/CVF International Conference on Computer Vision. 2019.
>
> [5] Zaken, Elad Ben, Yoav Goldberg, and Shauli Ravfogel. "BitFit: Simple Parameter-efficient Fine-tuning for Transformer-based Masked Language-models." Proceedings of the 60th Annual Meeting of the Association for Computational Linguistics (Volume 2: Short Papers). 2022.
>
> [6] Bojanowski, Piotr, et al. "Optimizing the Latent Space of Generative Networks." International Conference on Machine Learning. PMLR, 2018.
>
> &nbsp;

---

> > ### Comment · Reviewer_1qaw · 2022-11-21
> > **Concerns Addressed**
> >
> > I would like to thank the authors for their thorough response. I appreciate all the explanations and thoughtful answers to mine as well as other reviewers' comments and questions. Besides other clarifications, I really appreciate A3. I think that this is an interesting direction and I recommend this submission for ICLR.

---

> ### Author Response · Authors · 2022-11-18
> **Rebuttal from Authors of Paper5139 to Reviewer 1qaw (3/4)**
>
> ---
>
> > **Q3.** From Table 2 it appears that fine-tuning is very important. This is fine as long as we care about the performance on the novel tasks only. But once finetuned, the model could forget the training tasks. This makes the approach less significant from the point of continual learning. Could you evaluate the forgetting, i.e. what is the performance on the training tasks after finetuning on the novel tasks?
>
> **A3.** We appreciate the thoughtful comment. We would like to clarify that our model **does not forget** the training tasks during fine-tuning. This is because we freeze all model parameters except the image encoder's bias parameters during fine-tuning (Section 3.3); that is, all task-specific knowledge is included in the bias parameters of the image encoder. Therefore, the knowledge acquired from past tasks can be recalled without forgetting by keeping the corresponding bias parameters and switching to them whenever a past model is needed. We especially note that the size of bias parameters is fairly small (288 KB, which amounts to keeping about ~3 labeled images of 256x256 resolution for each task). This allows our model to retain past knowledge very efficiently by keeping the tuned bias parameters plus a few-shot support set, whose external memory requirement is far less compared to memory-based approaches in continual learning that keep hundreds of images [1, 2].
>
> While the continual learning setting is not our main focus, applying our method to a continual learning setting would be an interesting future direction. We appreciate the reviewer’s comment and will add the above discussion to the main text.
>
> [1] Bang, Jihwan, et al. "Rainbow memory: Continual learning with a memory of diverse samples." Proceedings of the IEEE/CVF Conference on Computer Vision and Pattern Recognition. 2021.
>
> [2] Wang, Zhen, et al. "Continual Learning With Lifelong Vision Transformer." Proceedings of the IEEE/CVF Conference on Computer Vision and Pattern Recognition. 2022.
>
> &nbsp;
>
> ---
>
> > **Q4.** Training for more tasks jointly is a harder job for the model but makes the dependency on the support set stronger. How important is that aspect for the proposed model? If you train on 5 tasks and test on 5 vs 8-2 vs 2-8, how does the performance change? It would be nice if you keep 2 novel tasks as fixed and then train the model on 2, 5 and 8 tasks and show how the performance on the two held-out tasks changes.
>
> **A4.** We appreciate the valuable comment and added the following discussion to the paper. Indeed, the amount of meta-training tasks is an important factor that can affect the performance of the universal few-shot learner. To verify this, following the reviewer’s suggestion, we fixed two test tasks (SS, SN) and trained our VTM with 2, 5, and 8 training tasks. To account for the potential variance that arises from the choice of training tasks, we choose multiple subsets of the training tasks of sizes 2 and 5, then report the mean and standard deviation. We summarize the results in the table below and Table 5 in Appendix C.1.4.
>
> |  Number of Training Tasks | SS (mIoU ↑) | SN (mErr ↓) |
> |:---:|:---:|:---:|
> | 2 | 0.3081 ± 0.0932 | 16.1029 ± 4.7710 |
> | 5 | 0.3702 ± 0.0029 | 12.7073 ± 0.2592 |
> | 8 | **0.4210** | **12.4863** |
>
> As expected, the performance consistently improves as we increase the number of training tasks.
>
> In addition, we kindly refer the reviewer to our additional experiment with incomplete training data (Appendix C.1.5), which shows the potential ability of our methods in more realistic settings where the training dataset is formed by a combination of different task-specific datasets. From these results, we expect that our model can further enhance its universality on few-shot learning by utilizing a combined training dataset of much more diverse tasks, which we leave as future work.
>
> &nbsp;

---

> ### Author Response · Authors · 2022-11-18
> **Rebuttal from Authors of Paper5139 to Reviewer 1qaw (2/4)**
>
> ---
>
> > **Q2.** The proposed architecture is not specifically designed for few-shot settings. In fact, if we compare the w/o Matching variant from Table 2 to the results of the baselines in Table 1, it does pretty well. This is good but also raises the question of how much the fast generalization is a result of the powerful architecture vs the multi-tasking meta-training. These additional experiments would shed light on this: (a) What is the performance of the proposed model if trained fully-supervised? Does it scale well when a lot of data is there? How does it compare to DPT? (b) How does out-of-the-box DPT work after few-shot finetuning?
>
> **A2.** We appreciate the valuable comment and added the following discussion to the paper. To understand the source of the fast generalization more clearly, we conducted additional experiments, including ones suggested by the reviewer.
>
> First, we further ablate two training procedures from our method, (1) BEiT pre-training and (2) episodic meta-learning. Specifically, we compare the following models:
>
> * **M1**: Randomly initialized DPT, 10-shot trained.
> * **M2**: DPT with BEiT pre-trained encoder, 10-shot fine-tuned.
> * **M3** (Ours w/o Matching): DPT with BEiT pre-trained encoder, multi-task trained with task-specific bias tuning, and then 10-shot fine-tuned.
> * **M4** (Ours): DPT with BEiT pre-trained encoder, meta-trained with task-specific bias tuning, and then 10-shot fine-tuned.
>
> We summarize the result in the table below and Table 4 in Appendix C.1.2, and also provide a qualitative comparison in Figure 8 in Appendix C.1.2.
>
> |   | SS (mIoU ↑) | SN (mErr ↓) | ED (RMSE ↓) | ZD (RMSE↓) | TE (RMSE↓) | OE (RMSE↓) | K2 (RMSE↓) | K3 (RMSE↓) | RS (RMSE↓) | PC (RMSE↓) |
> |:---:|:---:|:---:|:---:|:---:|:---:|:---:|:---:|:---:|:---:|:---:|
> | M1 | 0.0777 | 22.0718 | 0.2148 | 0.0865 | 0.0740 | 0.1772 | 0.0608 | 0.0602 | 0.2385 | 0.0434 |
> | M2 | 0.3070 | 15.4657 | 0.1490 | 0.0557 | 0.0700 | 0.1384 | **0.0573** | 0.0518 | 0.2021 | 0.0427 |
> | M3 | 0.3594 | 12.9554 | 0.1111 | 0.0388 | **0.0633** | 0.1014 | 0.0596 | 0.0502 | 0.1420 | 0.0410 |
> | M4 | **0.4210** | **12.4863** | **0.0728** | **0.0306** | 0.0634 | **0.0889** | 0.0625 | **0.0452** | **0.1063** | **0.0408** |
>
> First, as expected, we observe that DPT with naive 10-shot training (M1) fails to generalize to the test examples in most of the tasks, except for two 2D texture-related tasks (TE, K2). We conjecture that TE and K2 are “easy” cases in terms of few-shot learning, as they are defined as low-level computational algorithms on RGB images, while other high-level tasks require knowledge about semantics (SS) or 3D space (SN, ED, ZD, OE, K3, RS, PC).
>
> Second, we note that BEiT pretraining (M2) largely improves the few-shot generalization performance, allowing the model to produce coarse predictions of the dense labels. However, it still cannot capture object-level fine-grained details in many tasks.
>
> Third, we observe that multi-task training and few-shot adaptation, combined with an efficient parameter-sharing strategy of bias tuning (M3, M4), further improves the performance with a clear gap with M2 where the predictions are also qualitatively finer than M2’s.
>
> Finally, as discussed in Section 5.2.1, M4 still further improves over M3 with a clear gap. This shows that in a few-shot learning setting, our matching framework and episodic training are more effective than simple multi-task pretraining employed in M3.
>
> In summary, we may conclude that the fast generalization of Ours is benefitted from episodic training of various tasks followed by parameter-efficient few-shot adaptation as well as powerful pre-training of the encoder (BEiT).
>
> &nbsp;
>
> While the above experiment clarifies the powerful DPT architecture itself is not a major source of the fast generalization, following the reviewer’s suggestion (a), we also fine-tuned our VTM with full supervision of test tasks. For the fine-tuning, we used the same training dataset as the fully-supervised DPT and employed the episodic fine-tuning objective (Section 3.3). For evaluation, since providing the entire training data as the support set for the matching module is infeasible, we provide a random subset of the training data as the support set to the model. We provide more details of this fine-tuning and evaluation setup in the answer below (**A4**).
>
> We summarize the result in Figure 9 in Appendix C.1.3. In most tasks, our model consistently improves when more supervision is given. With full supervision at test tasks, our model performs slightly worse than the DPT baseline in seven tasks and performs better or similarly in the other three tasks. For the tasks where our model underperforms DPT under full supervision, we conjecture two potential causes: (1) the absence of direct input-output connection in our model, *i.e.*, the matching module serves as a bottleneck, and (2) negative transfer from meta-training tasks to test tasks.
>
> &nbsp;

---

> ### Author Response · Authors · 2022-11-18
> **Rebuttal from Authors of Paper5139 to Reviewer 1qaw (1/4)**
>
> ---
> > **Q1.** Could you please explain the technical differences between the w/o Matching baseline and DPT? Equivalently, if we add a support set on DPT, how different is this from the proposed model?
>
> **A1.** While Ours w/o Matching baseline (Section 5.2.1) adopts the encoder-decoder architecture of fully-supervised DPT (Section 5.2), it has three non-trivial technical modifications also shared with Ours (VTM) that make it more suited for universal few-shot learning.
>
> First, while DPT assumes the availability of full supervision of target tasks and employs one-stage fully-supervised training for each task, Ours w/o Matching employs two-stage training, where the first stage (training) involves multi-task learning from the training tasks, and the second stage (few-shot adaptation) involves few-shot fine-tuning on the unseen target task.
>
> Second, while DPT learns its entire parameters task-specifically, Ours w/o Matching involves extensive parameter sharing across tasks, and the only task-specific ones are the bias parameters of the encoder. Specifically, we tune the bias parameters for each task separately during training, and then at few-shot adaptation, a new set of initialized bias parameters are fine-tuned for each unseen task using the given support set.
>
> Third, while the supervised DPT baseline predicts all channels of a multi-channel task jointly, Ours w/o Matching adopts channel-wise decomposition and solves each channel of multi-channel tasks independently, thus being able to generalize to multi-channel tasks of an unseen number of channels as discussed in Section 2.
>
> We hope this clarifies the reviewer’s understanding and are happy to elaborate more if the reviewer wants any further clarifications. ​We will also revise the text in Section 5.2.1 to clarify the distinction between Ours w/o Matching and DPT.
>
>
> &nbsp;

---

### Comment · Program_Chairs · 2023-03-21
**Congratulations on winning the ICLR 2023 Outstanding Paper Award!!**

Hello authors,

Big congratulations! For more details about the award this year, please see https://blog.iclr.cc/2023/03/21/announcing-the-iclr-2023-outstanding-paper-award-recipients/

Best,

Mengdi, on behalf of ICLR 2023 PCs

---

### Decision · Program_Chairs · 2023-01-20

**Decision:**

Accept: notable-top-5%

**Justification For Why Not Higher Score:**

N/A

**Justification For Why Not Lower Score:**

The results clearly  show that the model can generalize few-shot across different tasks and outperform previous models.  All reviewers agree on the novelty of the work and its good empirical performance.



**Metareview: Summary, Strengths And Weaknesses:**

The paper proposes a model for few-shot dense image prediction tasks, i.e. tasks where the input is an image and the output is a per-pixel label map. The paper proposes a Transformer-based meta-learning approach where a support set of labelled images and a query image are jointly encoded and cross attended to predict the query label map through explicitly key-valuu attentions between query pixels and support pixels and their label embeddings. Most of the parameters on the model are shared, and only few of them are finetuned  for each new task. The results clearly  show that the model can generalize few-shot across different tasks and outperform previous models.  All reviewers agree on the novelty of the work and its good empirical performance.

**Note From Pc:**

if the above contains the word "oral" or "spotlight" please see: "oral" presentation means -> notable-top-5% and "spotlight" means -> notable-top-25%. As stated in our emails, we are disassociating presentation type from AC recommendations

**Summary Of Ac-Reviewer Meeting:**

N/A